

# Decoding the Architecture of Drought: SHAP-Enhanced Insights into the Climate Forces Reshaping the Sahel

Fabio Di Nunno[1], Mehmet Berkant Yıldız[1], Francesco Granata[1]

[1]Department of Civil and Mechanical Engineering (DICEM), University of Cassino and Southern Lazio, Cassino, 03043, Italy

*Correspondence to*: Francesco Granata (f.granata@unicas.it)

**Abstract.** The Sahel region faces increasing drought variability, driven by complex interactions between climatic indices and hydrological extremes. This study explores the correlation between the Standardized Precipitation Evapotranspiration Index (SPEI) and multiple climatic indices using trend analysis, cross-correlation, and an innovative SHAP-driven (SHapley Additive exPlanations) clustering approach. The Seasonal Kendall (SK) test identified statistically significant decreasing SPEI-12 trends

in 57.5% of the gridded data, particularly in the western (Senegal and Mauritania) and southeastern regions of the Sahel (South Sudan). In contrast, 19.3% of the data, primarily in the central-western Sahel (Burkina Faso and Niger), exhibited statistically significant increasing trends. Correlation analysis between SPEI-12 and climatic indices revealed strong negative relationships between SPEI and Global Mean Temperature (GMT, correlation coefficient up to -0.76) and Indo-Pacific Warm Pool (IPWP, -0.71), underscoring their role in drought intensification. Conversely, the Atlantic Multidecadal Oscillation (AMO, 0.40)

showed a positive correlation, emphasizing its influence on regional hydrology. Clustering delineated three distinct drought-prone regions, with Cluster C2, including the Sahel regions of Senegal, Mauritania, and Mali, (western Sahel) experiencing the most severe drought intensification (Z = -5.04). The SHAP-driven clustering approach, which incorporates a Machine Learning (ML) Random Forest (RF) model to classify data points into clusters, allowing the SHAP method to quantify the influence of each climatic variable on the clustering process, further highlights the dominant role of AMO and the North

Tropical Atlantic Index (NTA) in shaping regional drought dynamics. This study provides a novel framework integrating explainable AI into drought assessment, offering valuable insights for climate adaptation and water resource management in the Sahel.

Keywords: Drought, SPEI, Climatic Indices, Clustering, SHAP, Sahel.

# 1 Introduction

Drought represents one of the most critical environmental threats in the Sahel, where chronic water scarcity profoundly influences agricultural productivity, food security, and overall socio-economic stability (Nicholson 2013). The region's climate is characterized by pronounced interannual rainfall variability, with monsoonal precipitation confined to a brief seasonal window each year (Guilbert et al. 2024). However, the intensification of climate change is exacerbating this variability,





triggering more frequent, prolonged, and severe droughts that disrupt hydrological cycles and amplify water stress (Taylor et al. 2013). These extreme events not only diminish water availability but also accelerate land degradation, intensify desertification, and deplete essential water reserves, particularly groundwater, further jeopardizing the resilience of local communities (Cuthbert et al. 2019). Recent trends indicate a worsening of drought conditions in the Sahel, with rainfall deficits persisting for multiple years or even decades. The 1970s and 1980s witnessed some of the most devastating droughts on record,

triggering widespread famine, mass displacement, and long-term ecological degradation (Sarr et al. 2024). Although sporadic rainfall recovery has been observed in recent decades, its distribution remains highly erratic, often manifesting as short-lived, intense storms that generate surface runoff rather than effectively replenishing groundwater reserves (Panthou et al. 2014). Meanwhile, rising temperatures further aggravate drought severity by intensifying evapotranspiration, depleting soil moisture, and significantly reducing groundwater recharge potential (Gleeson et al. 2012). Consequently, the Sahel faces an increasingly

unpredictable hydrological future, where traditional water management strategies may no longer suffice to mitigate the impacts of prolonged dry periods.

    The socio-economic repercussions of these droughts are profound, affecting millions who rely on rain-fed agriculture and pastoralism for their livelihoods (MacDonald et al. 2012). As surface water sources become increasingly unreliable, communities are forced to extract groundwater at unsustainable rates, accelerating resource depletion (Hamdi et al. 2020). This

over-extraction, coupled with diminished recharge opportunities due to shifting precipitation patterns, raises serious concerns about long-term water security (Döll and Fiedler 2008). Furthermore, the intensifying scarcity of water resources has been linked to escalating social tensions, forced migration, and regional conflicts, underscoring the urgent need for adaptive and sustainable drought management strategies.

    A comprehensive assessment of drought impacts is essential for understanding its cascading effects on agriculture, ecosystems,

and communities, ultimately enabling more effective adaptation strategies and informed decision-making. Drought patterns are governed by a complex interplay of climatic variables that regulate atmospheric circulation, precipitation dynamics, and temperature anomalies. Large-scale climate drivers can exacerbate or prolong drought conditions, altering hydrological cycles, intensifying water deficits, and increasing the frequency of extreme events. Prolonged dry spells deplete soil moisture, reduce river discharge, and lower reservoir levels, severely compromising water availability for irrigation, drinking water supply, and

ecosystem stability. Without effective mitigation strategies, these disruptions could further threaten food security and socio-economic resilience in drought-prone regions like the Sahel. An in-depth investigation of drought variability in the Sahel necessitates a methodological framework capable of capturing both long-term trends and the complex interplay between drought conditions and large-scale climatic drivers. However, many of the methodologies traditionally employed for such analyses exhibit notable limitations. Conventional trend detection techniques, such as the Mann-Kendall (MK) test, are widely

applied to identify monotonic trends in hydroclimatic time series but are inherently less effective in detecting non-monotonic patterns, localized anomalies, and seasonal fluctuations, all of which are particularly relevant in a region characterized by pronounced climatic variability.





To overcome these constraints, the Seasonal Kendall (SK) test is employed in this study. This extension of the MK test accounts for intra-annual variations by stratifying data into seasonal components prior to trend detection, thereby mitigating biases introduced by periodic fluctuations (Hirsch and Slack, 1984). By applying the SK test to both the SPEI (Vicente-Serrano et al., 2010) and multiple climatic indices, a more accurate representation of long-term drought trends is achieved, ensuring that seasonality does not obscure or distort the underlying hydrological patterns.

Beyond trend analysis, an examination of drought variability requires a deeper understanding of its relationships with large-scale climate drivers, which regulate atmospheric circulation, precipitation dynamics, and temperature anomalies. Conventional correlation-based approaches, while informative, often fail to fully capture the temporal dependencies and potential nonlinearities that characterize these interactions. To address this issue, cross-correlation analysis is applied to quantify the strength and direction of associations between SPEI and key climatic indices. This approach enables the identification of dominant teleconnections influencing drought variability in the Sahel, providing a more nuanced, data-driven perspective on climate-hydrology linkages.

A critical yet frequently overlooked aspect of drought characterization concerns the identification of spatially homogeneous regions exhibiting consistent drought-climate relationships. Standard clustering techniques, such as K-means and Hierarchical clustering, offer a means of delineating regions with similar hydroclimatic characteristics but fail to provide insights into the relative influence of different climatic drivers on cluster formation. To overcome this limitation, an innovative SHAP-driven clustering framework is introduced, integrating SHAP with a RF classification model. While RF predicts cluster memberships, SHAP analysis quantifies the contribution of each climatic variable, thereby enhancing the interpretability of clustering outputs and offering a transparent framework for assessing the climatic controls on regional drought patterns. This approach constitutes a significant methodological advancement, as it not only identifies drought-prone zones but also elucidates the underlying climatic mechanisms that govern their formation.

This study introduces a pioneering multi-method framework that transcends conventional drought analysis by integrating trend detection, climate-drought interactions, and an interpretable ML-based clustering approach. Unlike traditional methodologies, which often rely on linear assumptions and overlook seasonal complexities, this approach employs the Seasonal Kendall test to capture nuanced hydrological trends, cross-correlation analysis to identify dominant climate drivers, and an innovative SHAP-driven clustering technique to reveal the underlying mechanisms governing spatial drought variability. By quantifying the contribution of individual climatic variables, this framework advances beyond conventional classification techniques, offering an unprecedented level of transparency and interpretability. These methodological advancements not only refine the characterization of drought dynamics in the Sahel but also provide a robust foundation for data-driven climate adaptation strategies, enabling more precise risk assessment and sustainable water resource management in vulnerable regions.



## 2 Materials and Methods

### 2.1 Study Area and Dataset

The Sahel (Figure 1) is a vast semi-arid region in Africa, extending across the continent from the Atlantic Ocean in the west to the Red Sea in the east. It forms a transitional zone between the arid Sahara Desert to the north and the more humid savannas to the south. Geographically, the Sahel spans parts of the main African Basins, including Senegal, Niger, Volta, Lake Chad and Nile rivers. Its topography is characterized by flat plains interspersed with rocky outcrops, dunes, and seasonal river systems. Morphologically, the Sahel is marked by sparse vegetation, dominated by drought-resistant grasses, shrubs, and

scattered trees, which are adapted to its harsh conditions. Soil types vary but are generally sandy and nutrient-poor, further constraining agricultural productivity. The climate of the Sahel is hot and dry, with a short and highly variable rainy season, typically lasting from June to September. Annual rainfall ranges between 100 and 600 millimeters, decreasing from south to north. The region is highly susceptible to prolonged droughts and erratic rainfall patterns, exacerbated by climate change. These conditions have profound impacts on water availability, agriculture, and the livelihoods of the predominantly rural

population, making the Sahel one of the most vulnerable regions to climatic and environmental stressors.

For this study, the SPEI gridded data from the Global SPEI Database (GSD) were utilized (details at https://spei.csic.es/). The GSD provides global coverage of SPEI data for the period from January 1901 to December 2023, with a spatial resolution of 0.5° and time scales ranging from 1 to 48 months. The spatial resolution of the GSD has been shown to be suitable for accurate drought analyses in regions with diverse climates, including the Northeastern United States (Krakauer et al., 2019), Iran

(Roushangar and Ghasempour, 2021), Somalia (Musei et al., 2021), Turkey (Danandeh Mehr and Attar, 2021), Southwest China (Sun et al., 2022), India (Vishwakarma et al., 2022), and Southern Italy (Di Nunno and Granata, 2023). This makes the GSD a valuable tool for studying drought in areas with limited or incomplete weather data. Figure 1 illustrates the study area and the 1335 SPEI gridded data points covering the Sahel region. The SPEI is advantageous as it incorporates the effects of evapotranspiration on drought severity. Specifically, it is based on the climatic water balance between precipitation (P) and

potential evapotranspiration (ETp). In the dataset used, ETp was computed using the well-established FAO-56 Penman-Monteith equation (Allen et al., 1998).

The SPEI can be calculated at various time scales, capturing both short- and long-term drought effects. For agricultural drought monitoring, 3- or 6-month scales are commonly used, whereas scales of 12 months or longer are applied for hydrological droughts (Tan et al., 2015). In this study, a 12-month time scale (SPEI12) was used. The choice of the 12-month timescale in

this analysis is grounded in the need to capture both seasonal and inter-annual variations in the Sahel's climate system, which are essential for understanding drought dynamics and water availability. The Sahel, characterized by a highly variable climate, experiences significant shifts between wet and dry periods. The 12-month timescale is particularly well-suited to account for these fluctuations, as it integrates both the seasonal rainfall patterns (which typically occur during the rainy season) and the cumulative effect of evapotranspiration, offering a more comprehensive measure of drought conditions over a yearly cycle.





Based on SPEI values, drought or wetness severity is classified from extreme drought (SPEI < -2) to extremely wet conditions (SPEI > 2).

Moreover, drought investigation faces significant challenges due to the complex, nonlinear, and dynamic nature of atmospheric processes. These intricacies hinder the accurate representation of spatial and temporal drought patterns, multi-scale interactions, and the influence of factors such as local topography and extreme climatic events. To enhance drought analysis,

the time series of various climate indices were integrated into the modeling framework. These indices offer valuable insights into large-scale climatic drivers and their interactions, which play a pivotal role in shaping drought dynamics. For example, shifts in the AMO significantly influence SPEI in the Sahel region, with warm AMO phases associated with increased precipitation and El Niño events linked to drought conditions (Okonkwo, 2014). From this perspective, the 12-month SPEI timescale allows for a clearer correlation with climatic indices. The 12-month period is long enough to capture the cumulative

effect of these global drivers, including any shifts in ocean temperatures or atmospheric circulation that might influence the region's climate over several seasons. By using this timescale, the SPEI effectively reflects the overall impact of global climate patterns on the Sahel's precipitation and evapotranspiration processes, allowing for stronger, more coherent correlations between the SPEI and these indices. This helps identify longer-term trends and anomalies, which are critical for understanding climate variability and drought conditions in the region

It should also be noted that, although the analysis included 31 climatic indices, the historical time series from 1951 to 2018 were suitable for this study. This period offered a comprehensive view of long-term climatic trends and variability, facilitating a robust analysis despite the large number of indices considered. Spanning both phases of climatic stability and significant change, the 1951–2018 timeframe provided a rich dataset for identifying correlations and understanding the long-term behavior of climatic drivers.

Table 1 summarizes the climate indices considered, emphasizing their relevance to drought variability. In the context of SPEI analysis, these indices are crucial for understanding temporal patterns and variations.




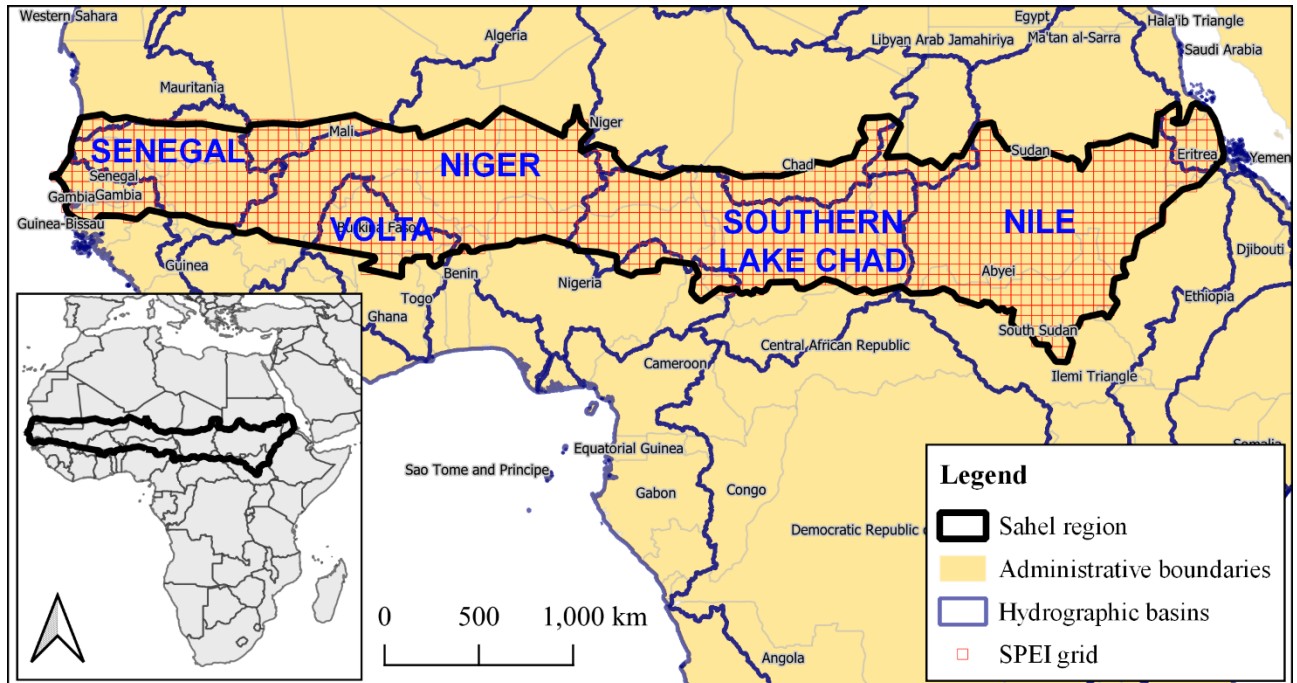

**Figure 1: Location of the selected SPEI grid in the Sahel region.**

**Table 1: Definition of the climatic indices.**

| Climate index | Abbr. | Definition |
|---|---|---|
| Atlantic Meridional Mode | AMM | The Atlantic Meridional Mode describes north-south asymmetries in sea surface temperatures across the tropical Atlantic Ocean. In its positive phase, North Atlantic temperatures warm while the South Atlantic cools, shifting tropical rainfall northward and increasing tropical cyclone activity. In its negative phase, the pattern reverses, with rainfall shifting southward and reduced storm activity. AMM is a key driver of Atlantic climate dynamics and precipitation patterns. |
| Atlantic Multidecadal Oscillation | AMO | The Atlantic Multidecadal Oscillation refers to natural variations in North Atlantic Ocean sea surface temperatures that occur over periods of 20 to 40 years. In its positive phase, North Atlantic temperatures are above average, leading to hotter summers along the eastern U.S., increased hurricane activity in the tropical Atlantic, and enhanced rainfall in Africa. In its negative phase, cooler Atlantic temperatures are associated with weaker hurricane activity, drought in Africa's Sahel region, and cooler, wetter summers in Europe. The AMO plays a significant role in shaping global climate systems and regional weather patterns, particularly in the North Atlantic region. |
| Arctic Oscillation | AO | The Arctic Oscillation is a climate pattern defined by atmospheric pressure differences between the Arctic and mid-latitudes, significantly influencing winter weather in the Northern Hemisphere. In its positive phase, low pressure strengthens over the Arctic, keeping the jet stream tightly confined and cold air locked in the polar region, resulting in milder winters in mid-latitudes. In its negative phase, high pressure dominates the Arctic, weakening the jet stream and allowing cold Arctic air to spill into mid-latitudes, causing harsher winter conditions. The AO plays a critical role in shaping weather and climate patterns across Europe, Asia, and North America. |
| Berkeley Earth | BEST | The BEST is climate metrics developed by the Berkeley Earth Surface Temperature project, focusing on global and regional surface temperature trends. These indices provide a comprehensive |





| Surface Temperature | | view of temperature anomalies across different spatial and temporal scales, helping to track climate change. |
|---|---|---|
| Caribbean Index | CAR | The CAR index represents climate variability in the Caribbean region, often focusing on sea surface temperature (SST) and atmospheric patterns. It is used to study tropical cyclone activity, precipitation changes, and regional climate impacts. |
| Eastern Pacific Oscillation Index | EPO | EPO refers to the variation of atmospheric pressure anomalies in the eastern North Pacific. The positive phase is dominated by low pressure in Alaska and western Canada, while the negative phase is dominated by high pressure. This pattern affects the jet stream and temperature and precipitation patterns in North America. |
| Greenland Blocking Index | GBI | The Greenland Blocking Index measures the strength and persistence of high-pressure systems over Greenland. In its positive phase, strong high pressures can lead to cold air outbreaks in the North Atlantic and colder winters in Europe. In its negative phase, strengthened westerly bring milder conditions to Europe. |
| Global Mean Temperature | GMT | Global Mean Temperature measures the overall warming or cooling trend of the atmosphere and oceans. Rising temperatures are typically linked to increased greenhouse gas concentrations, resulting in effects like glacial melt and sea level rise. Cooler periods indicate more stable or cooling climate conditions. |
| Indo-Pacific Warm Pool | IPWP | The Indo-Pacific Warm Pool is a vast tropical region in the Indian and Western Pacific Oceans where sea surface temperatures typically exceed 28°C. It contains the warmest ocean surface temperatures on Earth and plays a critical role in global atmospheric circulation, including monsoon systems and tropical cyclone formation. In its positive phase (warming), convective activity and rainfall increase in the region, influencing tropical weather systems and global climate patterns. In its negative phase (cooling), reduced rainfall and weakened monsoon systems can occur. |
| The North Atlantic Oscillation | NAO | The North Atlantic Oscillation (NAO) index is based on the surface sea-level pressure difference between the Subtropical (Azores) High and the Subpolar Low. The positive phase of the NAO reflects below-normal heights and pressure across the high latitudes of the North Atlantic and above-normal heights and pressure over the central North Atlantic, the eastern United States and western Europe. The negative phase reflects an opposite pattern of height and pressure anomalies over these regions. Both phases of the NAO are associated with basin-wide changes in the intensity and location of the North Atlantic jet stream and storm track, and in large-scale modulations of the normal patterns of zonal and meridional heat and moisture transport, which in turn results in changes in temperature and precipitation patterns often extending from eastern North America to western and central Europe. |
| The North Atlantic Oscillation (Jones) | NAO (Jones) | The NAO index, as defined by Jones (1981), is based on the difference in sea-level pressure between two locations: the Azores High (Subtropical High) and the Icelandic Low (Subpolar Low). This index quantifies the strength and phase of the NAO, which oscillates between positive and negative phases. |
| Niño-1.2 | - | The Niño-1+2 index covers a smaller area in the eastern equatorial Pacific (80°W-90°W, 10°S-0°), representing the region where El Niño and La Niña events typically originate. In its positive phase (El Niño), sea surface temperatures in this area rise above average, causing increased rainfall and flood risks along the northwestern coast of South America. In its negative phase (La Niña), cooler-than-average temperatures prevail, leading to drought in the region while supporting beneficial cold-water upwelling for marine life and fisheries. This region plays a crucial role in global climate dynamics. |
| Niño 3 | - | The Niño-3 index measures sea surface temperatures in the eastern equatorial Pacific (150°W-90°W, 5°S-5°N) and is a key indicator of El Niño and La Niña events. In its positive phase (El Niño), sea surface temperatures are higher than average, leading to increased rainfall along the western coast of South America and drought in the Asia-Pacific region. In its negative phase (La Niña), cooler-than-average temperatures prevail, resulting in heavy rainfall and storm activity in |





| | | the Asia-Pacific region and drought in western South America. This index significantly influences global atmospheric circulation and tropical rainfall patterns. |
|---|---|---|
| Niño-3.4 | - | The Niño-3.4 index covers a central equatorial Pacific region (120°W-170°W, 5°S-5°N) and is one of the most prominent indicators of El Niño and La Niña events. In its positive phase (El Niño), sea surface temperatures rise above average, causing increased rainfall along the South American coasts and droughts with heatwaves in the Asia-Pacific region. In its negative phase (La Niña), cooler-than-average sea surface temperatures lead to heavy rainfall and flood risks in the Asia-Pacific region, while drought conditions dominate South America. Niño-3.4 has widespread impacts on global climate, influencing atmospheric circulation and tropical weather patterns. |
| Niño-4 | - | The Niño-4 index measures sea surface temperature variations in a central Pacific region (160°E-150°W, 5°S-5°N) during El Niño and La Niña events. In its positive phase (El Niño), sea surface temperatures are warmer than average, leading to increased convective activity in the tropical western Pacific, which influences atmospheric circulation and monsoon patterns across the Pacific. In its negative phase (La Niña), cooler-than-average temperatures dominate, associated with drought in the western Pacific and enhanced rainfall in the central Pacific. The Niño-4 index is crucial for understanding tropical atmospheric circulations and regional climate dynamics. |
| Northern Oscillation Index | NOI | The Northern Oscillation Index measures atmospheric pressure differences in the tropical Pacific, based on the sea level pressure between Tahiti in the eastern tropical Pacific and Darwin in the western subtropical Pacific. In its positive phase, high pressure dominates the eastern Pacific and low pressure the western Pacific, leading to drought in the eastern Pacific and wetter conditions in the western Pacific. In its negative phase, the pattern reverses. The NOI is a key tool for understanding the impacts of tropical climate phenomena like the El Niño-Southern Oscillation |
| North Pasific Index | NP | The North Pacific Index measures the average sea level pressure over the North Pacific Ocean, often representing the strength of the Aleutian Low-Pressure System. In its positive phase, the Aleutian Low intensifies, leading to increased rainfall in western North America and cooler sea surface temperatures in the eastern Pacific. In its negative phase, the pressure system weakens, resulting in drier conditions in western North America and warmer sea surface temperatures in the Pacific. The NP Index is essential for understanding atmospheric circulation patterns in the Pacific and their impact on North American climate. |
| North Tropical Atlantic SST Index | NTA | The NTA index represents sea surface temperature (SST) anomalies in the North Tropical Atlantic region. It is a significant indicator influencing tropical cyclone activity, precipitation patterns, and atmospheric circulation over the Atlantic Ocean. |
| Oceanic Niño Index | ONI | The ONI is an index used to characterize El Niño and La Niña events, measuring the three-month moving average of sea surface temperature anomalies in the 3.4 region of the tropical Pacific Ocean. The ONI is an indicator of the ENSO (El Niño-Southern Oscillation) cycle, which influences global climate models. |
| Pasific Decadal Oscillation | PDO | The Pacific Decadal Oscillation refers to multi-decadal changes in sea surface temperatures and atmospheric pressure across the North Pacific Ocean. In its positive phase, warm sea surface temperatures dominate the western Pacific, while cooler temperatures prevail in the eastern Pacific, leading to increased rainfall along the western coast of North America and warmer conditions in Alaska. In its negative phase, this pattern reverses, resulting in droughts in western North America and reduced marine ecosystem productivity along Pacific coasts. PDO significantly influences long-term climate variability, impacting agriculture, fisheries, and water resources. |
| Pacific Meridional Mode | PMM | The Pacific Meridional Mode is a climate pattern defined by interactions between sea surface temperatures and atmospheric winds in the tropical Pacific Ocean, potentially triggering events like the El Niño-Southern Oscillation (ENSO). In its positive phase, sea surface temperatures in the tropical Pacific increase, enhancing convective activity and rainfall across the Pacific. In its negative phase, cooler sea surface temperatures prevail, leading to calmer tropical conditions and |



| | | |
|---|---|---|
| | | reduced rainfall. PMM plays a significant role in shaping tropical climate systems and global climate patterns. |
| Pacific–North American Pattern | PNA | The Pacific-North American (PNA) pattern is one of the most prominent modes of atmospheric variability in the Northern Hemisphere. It describes a recurring and persistent pattern of atmospheric pressure anomalies over the North Pacific Ocean and North America. The PNA index is calculated using standardized geopotential height anomalies at 500 hPa at specific geographic locations. A pattern of air pressure anomalies—departures from the long-term average—at four locations over the Pacific Ocean and North America correlate with regional temperature and precipitation anomalies across North America. This pattern, known as the Pacific-North American teleconnection pattern or PNA, influences regional weather by affecting the strength and location of the East Asian jet stream, and subsequently, the weather it delivers to North America. |
| Quasi-Biennial Oscillation | QBO | The QBO is a climate phenomenon characterized by the oscillation of easterly and westerly winds in the tropical stratosphere with a periodicity of approximately 28-30 months. In its positive phase (westerly winds), the formation and intensity of tropical cyclones, particularly in the Pacific and Atlantic, tend to increase. In its negative phase (easterly winds), tropical cyclones weaken, and changes in stratospheric ozone distribution occur. The QBO regulates interactions between the tropical stratosphere and troposphere and influences global climate patterns. |
| Sahel Precipitation | Sahel P | ahel Precipitation refers to the annual rainfall in Africa's Sahel region (south of the Sahara Desert) and is influenced by tropical Atlantic Sea surface temperatures and atmospheric circulation. Positive phases (increased rainfall) improve agriculture and water resources, while negative phases (drought) lead to famine and heightened socio-economic impacts. |
| Southern Oscillation Index | SOI | The SOI is calculated based on the difference in air pressure anomalies between two locations in the tropical Pacific: Tahiti (central Pacific) Darwin, Australia (western Pacific). The formula for the SOI involves standardizing the pressure difference to account for seasonal variations and long-term trends. |
| Solar Flux | - | Solar flux refers to the amount of energy from the Sun reaching Earth and is often used as an indicator of solar activity cycles. Increased solar flux can influence Earth's energy balance, leading to warming and changes in atmospheric circulation. During periods of low solar flux, cooling effects and more stable climate conditions may be observed. |
| Tropical Northern Atlantic Index (TNA) | TNA | The Tropical Northern Atlantic Index measures sea surface temperature anomalies in the tropical North Atlantic region (5°N-25°N, 15°W-60°W). In its positive phase, sea surface temperatures are warmer than average, leading to increased rainfall along the West African coast and enhanced tropical cyclone activity. In its negative phase, cooler sea surface temperatures are associated with reduced tropical rainfall and weaker cyclone activity. TNA plays a crucial role in understanding climate dynamics and weather patterns in the tropical Atlantic |
| Trans Nino Index | TNI | The Trans-Niño Index is used to analyze spatial shifts of El Niño and La Niña events, defined by the difference in sea surface temperatures between the eastern tropical Pacific (Niño-1+2 region) and the central tropical Pacific (Niño-4 region). In its positive phase, El Niño effects shift eastward, resulting in stronger temperature anomalies and increased rainfall in the eastern Pacific. In its negative phase, La Niña effects shift eastward, leading to cooler temperatures and drought in the eastern Pacific. The TNI is a crucial tool for understanding the spatial and temporal variability of El Niño-Southern Oscillation (ENSO) events. |
| Tropical Southern Atlantic Index | TSA | The Tropical Southern Atlantic Index measures sea surface temperature anomalies in the tropical South Atlantic region (0°-20°S, 10°E-30°W). In its positive phase, warmer-than-average sea surface temperatures can lead to increased rainfall along the eastern coast of South America and shifts in the Atlantic Hadley circulation. In its negative phase, cooler temperatures are associated with drought and reduced tropical convection activity. TSA is crucial for understanding climate events and atmospheric circulation in the tropical South Atlantic. |
| Tropical Western | WHWP | The Tropical Western Hemisphere Warm Pool encompasses the Caribbean, Gulf of Mexico, and eastern tropical Pacific, where sea surface temperatures typically exceed 28°C. This warm pool |





| Hemisphere warm pool | | significantly influences the formation and intensity of tropical cyclones, as well as atmospheric circulation and rainfall patterns. In its positive phase, increased temperatures lead to more intense rainfall and stronger storms in the Caribbean and surrounding regions. In its negative phase, cooler sea surface temperatures are associated with reduced rainfall and weaker tropical cyclone activity. |
|---|---|---|
| West Pasific Index | WPI | The West Pacific Index measures atmospheric pressure differences in the tropical and subtropical western Pacific Ocean. In its positive phase, a strong high-pressure system dominates the region, potentially weakening Asian monsoons and reducing tropical cyclone activity. In its negative phase, low-pressure systems prevail, enhancing Asian monsoons and increasing storm activity across the Pacific. The West Pacific Index is essential for understanding the dynamics of tropical climate systems and atmospheric circulation. |

## 2.2 Modeling procedure

To better understand the complex dynamics of drought fluctuations, this study adopted a comprehensive analytical framework that incorporated the SK test, along with correlation and clustering analyses. The SK test was selected due to its ability to account for seasonal variations in SPEI and climatic indices time series, addressing the shortcomings of conventional methods like the MK test, which often fail to consider the seasonality inherent in groundwater dynamics. In the case of SPEI, a negative Z-value obtained from the SK test signals a decline typically linked to drought conditions, whereas a positive Z-value indicates an increase often associated with wetter climatic scenarios. In this study, the SK test was conducted with a strict 95% confidence level (p-value ≤ 0.05) to enhance the reliability of the identified trends. For further details, refer to the foundational works of Hirsch and Slack (1984) and a recent application by Di Nunno et al. (2023).

Then, to gain a deeper understanding of how climatic indices influence drought conditions, this study analyzed the correlation between SPEI and various climatic indices. This analysis is crucial because climatic indices play a significant role in shaping drought dynamics. By examining these relationships, it becomes possible to identify the key drivers behind drought variability and their relative contributions over time. Such insights are essential not only for advancing scientific knowledge but also for informing targeted mitigation and adaptation strategies, particularly in regions where hydrological cycles are highly sensitive to climatic fluctuations.

Furthermore, by focusing on the climatic indices most strongly correlated with the SPEI, a clustering analysis was performed to divide the Sahel into homogeneous regions based on the correlation between SPEI and these climatic indices. This approach is entirely innovative, as it allows for the identification of distinct zones where specific climatic factors have a pronounced influence on drought dynamics. By delineating these homogeneous regions, this methodology provides valuable insights into localized drought drivers and their variability across the Sahel.

In general, clustering is the process of classifying a large dataset into a smaller number of groups, or clusters. Data within the same cluster shares common features, while data from different clusters exhibit some degree of heterogeneity (Barton et al. 2016). In this study, two widely recognized clustering algorithms, K-means and Hierarchical clustering, were preliminarily tested to divide the study area into homogeneous regions with shared characteristics. These algorithms have been previously applied in drought analyses to identify such regions. A detailed description of both algorithms can be found in the work of Di Nunno and Granata (2023).



It is important to note that the optimal clustering algorithm and the number of clusters are not predetermined. To address this, the Silhouette technique was applied. This widely recognized method is used for interpreting and validating the consistency of clustering results, offering a measure of how well an object fits within its assigned cluster compared to other clusters (Shutaywi and Kachouie 2021). The Silhouette score ranges from -1, indicating that clusters are poorly assigned, to 1, signifying well-

separated and distinct clusters. A score close to 0 suggests that the separation between clusters is negligible or ambiguous. However, this criterion does not allow for a clear assessment of the impact of each climatic index on the clustering process. To address this limitation, an innovative approach in the field of drought analysis was introduced, leveraging SHAP analysis in combination with an RF model. RF was selected due to its robustness in handling complex, high-dimensional datasets and its ability to capture non-linear relationships between variables. This ML algorithm excels in managing diverse and noisy data,

common in climatic and environmental studies, by using an ensemble of decision trees to generate more accurate predictions and reduce the risk of overfitting.

Although SHAP is primarily used for supervised models, it can also be applied to clustering by employing a classification model to predict the assigned clusters. In this context, RF is used to classify the data points into clusters, and SHAP analysis is then applied to evaluate the importance of each climatic variable in determining the cluster assignments. This SHAP-Driven

Clustering, unveiling the explainable clustering process, offers a more transparent understanding of how each variable influences the clustering and provides a clear measure of the relative significance of each variable. By employing this method, the analysis not only gains enhanced interpretability but also fosters a more nuanced understanding of the climatic factors driving the clustering. This, in turn, improves the precision of the overall analysis and supports more informed decision-making strategies for addressing climate-related challenges.





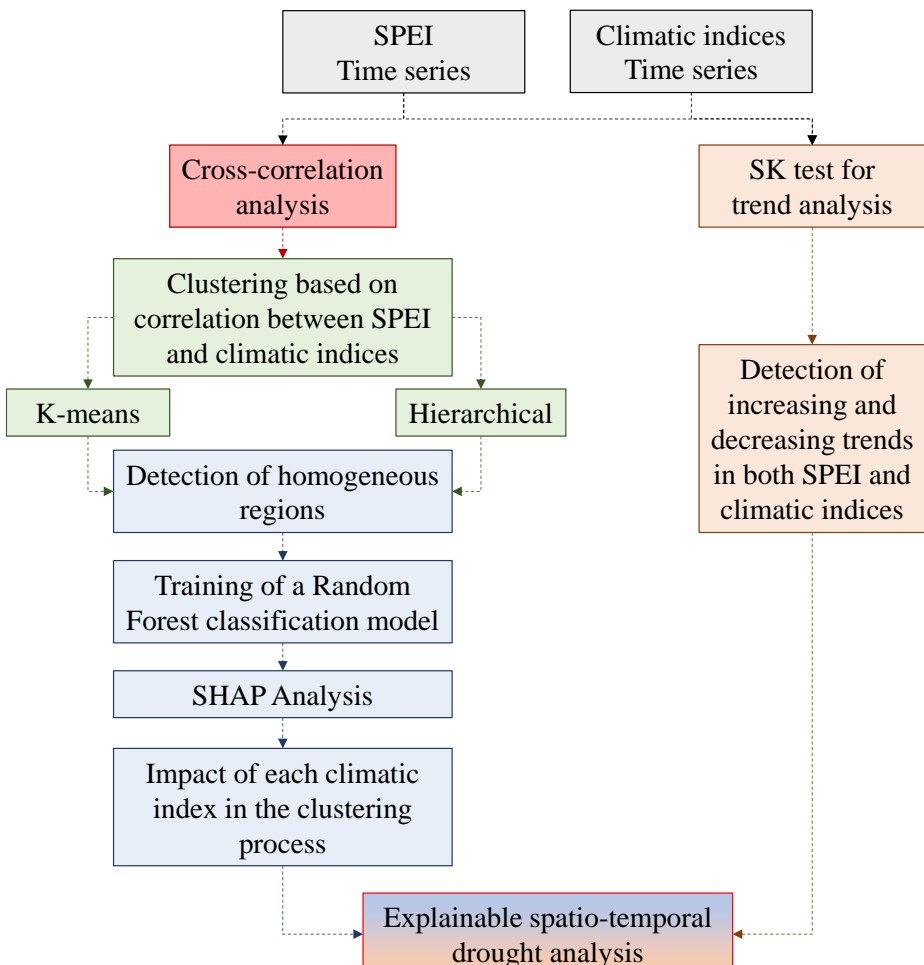

**Figure 2: Modeling procedure.**

## 3 Results

### 3.1 Trend analysis

The SK test analyzed SPEI-12 trends across the Sahel, highlighting patterns of drought and wetness (Figure 3a). Results showed that Z-values between -1.96 and 1.96 indicate no significant trends at the 95% confidence level, while values outside this range denote statistically significant trends. Significant trends were identified in various regions. Specifically, 57.5% of the cells exhibited statistically significant decreasing trends, while 19.3% showed statistically significant increasing trends. The remaining 23.2% displayed increasing or decreasing trends that were not statistically significant. Statistically significant decreasing trends were observed in the western Sahel, spanning from Senegal's Atlantic Coast and Gambia to Mali, as well as in the southeastern region, including South Sudan. Another area experiencing marked increases in drought is the northern-





central Sahel, particularly in Chad. These findings should be potentially linked to reduced rainfall and rising temperatures. On the other hand, the Sahel region between Burkina Faso and Nigeria exhibited increasing SPEI-12 trends, indicating a tendency toward wetter conditions. These wetter tendencies highlight the heterogeneous nature of climatic changes across the Sahel, where some areas may benefit from increased rainfall while others face mounting water stress.

The SK test was performed also for the climatic indices (Figure 3b). The predominance of statistically significant increasing trends, particularly for the IPWP (Z = 27.83) and GMT (Z = 28.70), underscores the substantial role of global warming and oceanic heat distribution in shaping regional climate dynamics. These trends align with broader observations of rising sea surface temperatures and global temperature anomalies, which are known to influence precipitation patterns and drought intensification in the Sahel. Conversely, the statistically significant decreasing trends in three indices, TNI (Z = -7.83), Solar

Flux (Z = -3.18) and Sahel P (Z = -4.13), indicate specific climatic factors contributing to regional drying. The decline in the TNI suggests changes in the tropical atmospheric circulation that may reduce moisture transport to the Sahel. Similarly, a decrease in Solar Flux could reflect reduced solar radiation reaching the surface, potentially linked to increased atmospheric aerosols or cloud cover, further influencing regional precipitation. The downward trend in Sahel P directly captures the reduction in rainfall over the region, consistent with the observed intensification of drought.

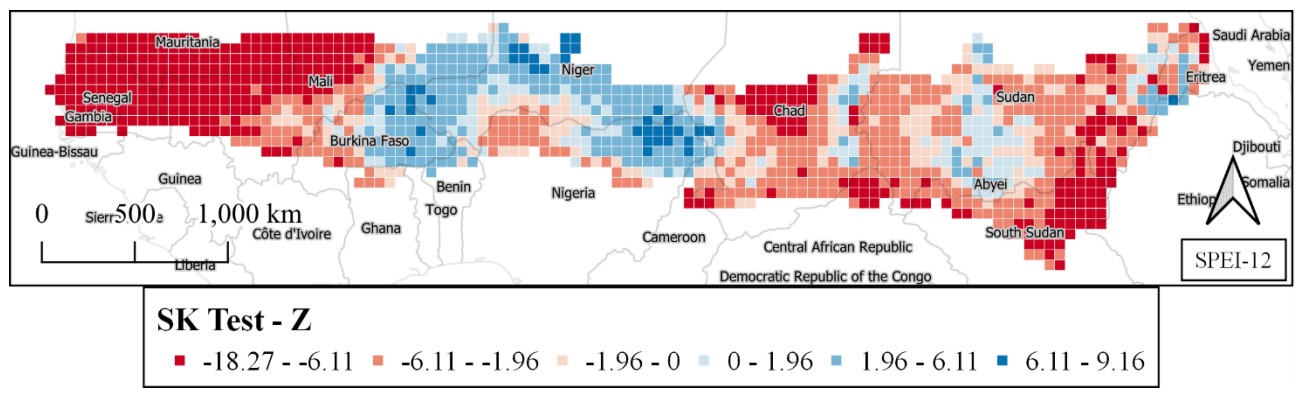


(a)





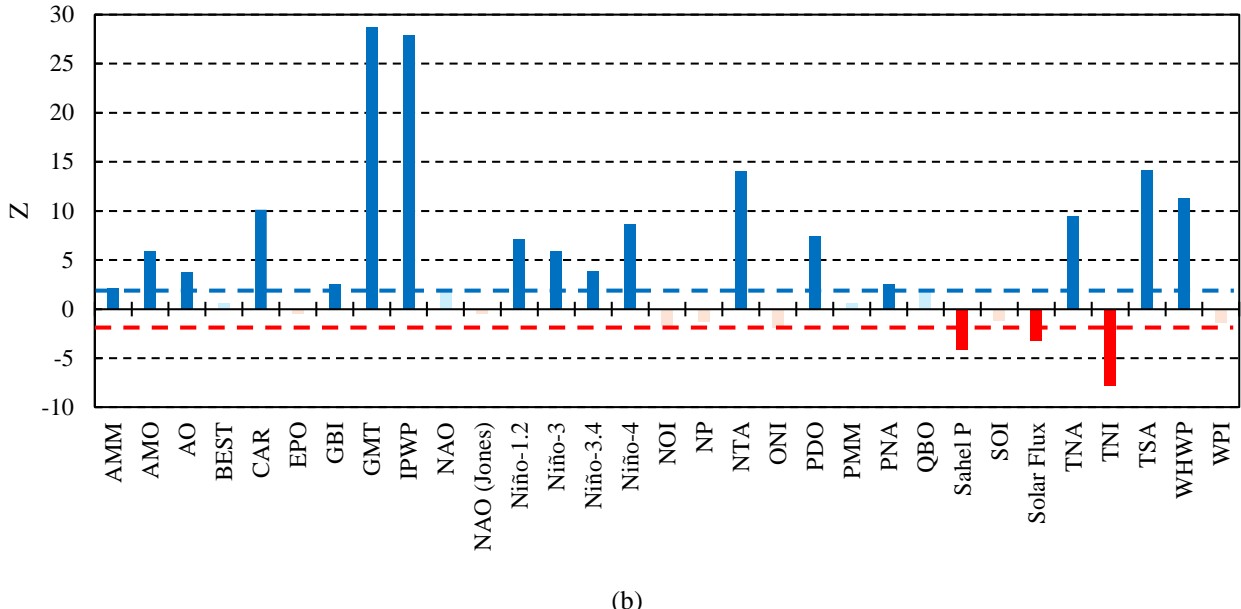

(b)

**Figure 3: Z parameter of the SK test: SPEI-12 map (a); climatic indices, with the blue and red dashed lines that indicate the statistically significant trend (Z = ± 1.96 – p-value = 0.05).**

### 3.2 Cross-correlation analysis

The cross-correlation analysis between the SPEI-12 and the different climatic indices was conducted to evaluate the influence of large-scale atmospheric and oceanic patterns on drought variability and intensity. This approach enables a deeper understanding of how global phenomena modulate Sahel's regional hydrological extremes.

Figure 4 provides the maps of the correlations between SPEI-12 gridded data and the most correlated climatic indices, while Figure 5 reports a combined box and violin plots representation of the correlations for all climatic indices. In addition, Tables S1 and S2 (see supplementary material) report the correlation coefficient and the p-value, respectively, computed between climatic indices. Finally, Table S3 provides the average, maximum, minimum values, and standard deviation of the correlations calculated between SPEI-12 gridded data and climatic indices.

The cross-correlation analysis between SPEI-12 and various climatic indices for the Sahel region reveals a complex interplay of global atmospheric and oceanic drivers on regional drought variability. The AMM (Interquartile range – IQR = 0.16) showed a moderate positive influence, with a mean correlation of 0.12 and peaks reaching 0.39. This suggests that the AMM's modulation of sea surface temperatures in the Atlantic plays a significant role in shaping precipitation patterns in the Sahel. Similarly, the AMO displayed a wide range of correlation values, from -0.59 to 0.40, with a modest mean of 0.06. Specifically,

the western regions of the Sahel, from Senegal's Atlantic coast to western Mali, exhibited a positive correlation with the SPEI-12. However, the strongest positive correlations were observed in the Sahel region along the border between Chad and Sudan. In contrast, the central-western region, including countries like Burkina Faso and Niger, showed a negative correlation. Figure





6 presents a detailed analysis of the correlation between SPEI-12 for Cell 2042 and the AMO. Among all analyzed locations, Cell 2042, located in the aforementioned border region between Chad and Sudan, exhibited the highest positive correlation
with the AMO index. This relationship is reflected in the long-term trends, where both SPEI-12 and AMO remained predominantly positive from the 1950s to the 1970s, turned negative from the 1970s to the late 1990s, and shifted positive again from the late 1990s to the present. This region, significantly affected by the oscillations of the Intertropical Convergence Zone (ITCZ), is characterized by highly seasonal rainfall, primarily governed by the West African Monsoon, and is highly sensitive to fluctuations in sea surface temperatures. The strong correlation with AMO suggests that warmer North Atlantic
conditions enhance monsoonal precipitation, while cooler phases contribute to drought conditions.

The GMT (IQR = 0.14) and IPWP (IQR = 0.13) indices exhibited strong negative correlations, reaching values of -0.76 and -0.71, respectively. These results underscore the adverse effects of warming in these regions, likely to intensify evapotranspiration and reduce soil moisture availability in the Sahel. The significant influence of GMT further highlights the overarching impact of global warming on regional hydrological cycles. Figure 7 presents a detailed analysis of the correlation
between SPEI-12 for Cell 2319 and the GMT. Among all analyzed locations, Cell 2319 exhibited the strongest negative correlation with the GMT index. This relationship is evident in the opposing trends of SPEI-12 and GMT: SPEI-12 was predominantly positive from 1950 to the late 1990s and negative from 2000 to the present, whereas GMT showed an inverse pattern, remaining mostly negative until the late 1990s before turning positive from 2000 onward. Cell 2319 is situated at the northern boundary of the Sahel in central Sudan, bordering the hyper-arid Sahara Desert. This region is highly sensitive to
climate variability, as it marks the transition between semi-arid and arid conditions. Rainfall is scarce and primarily influenced by the northward penetration of the West African Monsoon, which is highly susceptible to global temperature shifts. The strong negative correlation with GMT suggests that global warming has exacerbated aridity in this zone, likely by reducing monsoonal rainfall and intensifying evapotranspiration.

In contrast, the Sahel P (IQR = 0.08) index stood out with a maximum correlation of 0.35, reinforcing its role as a local climatic
driver that directly reflects rainfall conditions in the region. Meanwhile, indices such as the AO (IQR = 0.05) and NAO (IQR = 0.06) exhibited weaker correlations, indicating that while these global-scale patterns might influence the Sahel indirectly, their direct impact on drought variability is limited or less consistent.

The variability in correlations, as indicated by standard deviations (see Table S3), was notable across indices. For instance, the AMO (Dev.st = 0.16) and CAR (Dev.st = 0.14) reflected significant spatial and temporal heterogeneity in their relationships
with SPEI-12, whereas indices like the GBI and AO showed much lower variability (Dev.st < 0.05), suggesting more stable but weaker connections. These findings emphasize that while certain indices like AMO are closely tied to sub-regional drought dynamics, others like GMT and IPWP reveal broader, systemic influences linked to global warming.



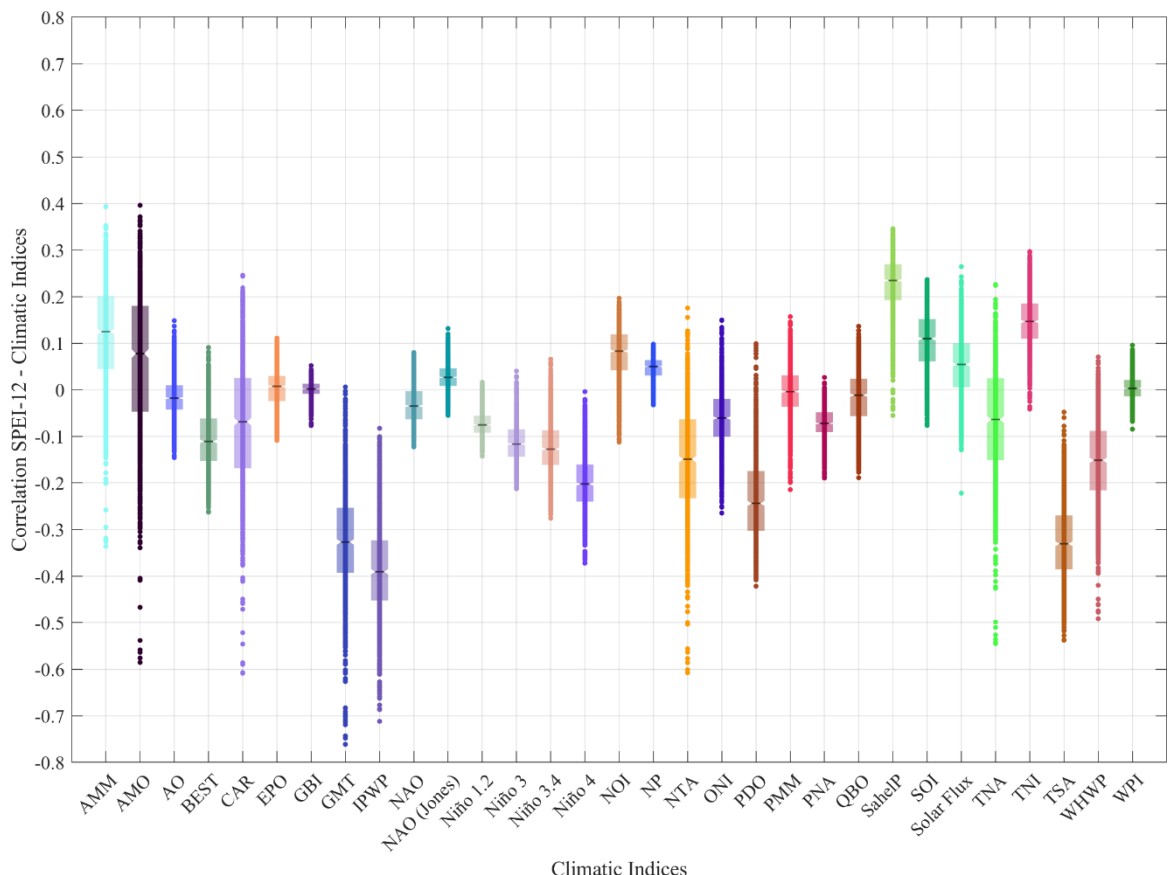

**Figure 4: Box plot representation of the correlations between SPEI-12 and climate indices.**





**SPEI-12 - CLIMATIC INDICES - Correlation coefficient (r)**

■ -1 - -0.75   ■ -0.75 - -0.5   ■ -0.5 - -0.25   -0.25 - 0   0 - 0.25   ■ 0.25 - 0.5   ■ 0.5 - 0.75   ■ 0.75 - 1






**Figure 5. Maps of the correlations between SPEI-12 gridded data and the most correlated climatic indices (continue).**



**Figure 6. Correlation analysis between SPEI-12 for Cell 2042 and AMO.** The figure presents the time series of AMO and SPEI-12 for Cell 2042, located at the border between Chad and Sudan. Additionally, it includes a scatter plot illustrating their relationship on both a monthly scale and a five-year mean scale.





**Figure 7.** Correlation analysis between SPEI-12 for Cell 2319 and GMT. The figure presents the time series of AMO and SPEI-12 for Cell 2319, located at the Sahel's border in Central Sudan. Additionally, it includes a scatter plot illustrating their relationship on both a monthly scale and a five-year mean scale.




### 3.3 Clustering

A preliminary analysis was conducted to determine the more suitable algorithm between K-means and Hierarchical clustering. The analysis considered a range of cluster numbers from 3 to 8. However, for the sake of brevity, only the results for the optimal number of clusters, identified as 3 based on the highest mean Silhouette Scores (see Figure 8a), are presented here. As inputs for the clustering, the most correlated climatic indices, whose correlations with SPEI-12 are depicted in Figure 5, were considered: AMM, AMO, CAR, GMT, IPWP, NTA, PDO, Sahel P, TNA, TSA and WHWP.

The minimum, mean, and maximum Silhouette Scores for the K-means and Hierarchical clustering algorithms are reported in Figure S1. K-means consistently achieved higher Silhouette Scores compared to Hierarchical clustering across all clusters. For K-means, the minimum, mean, and maximum scores ranged from 0.47 (C1) to 0.50 (C2), 0.58 (C1) to 0.61 (C3), and 0.64 (C1) to 0.66 (C3), respectively. In contrast, Hierarchical clustering exhibited negative minimum values, ranging from -0.47 (C1) to 0.21 (C3), along with lower mean scores, from 0.19 (C1) to 0.29 (C2), and maximum values, from 0.46 (C1) to 0.50 (C2). Therefore, the subsequent discussion focuses on the clustering analysis performed using the K-means algorithm, which has been considered and examined in detail.

Cluster C1 (blue circles in Figure 8b), covered most of the central-western Sahel, corresponding to part of Niger and Burkina Faso. Cluster C2 (red circles in Figure 8b) covered most of the western Sahel, including the Sahel regions of Senegal, Mauritania, and Mali, except for the southwestern portion, corresponding to Gambia and Southern Senegal, which falls under Cluster C3. Other parts of the Sahel also fall within Cluster C2, such as some central areas of the Sahel and the easternmost part of the Sahel, including a part of Sudan, South Sudan and Eritrea, which is shared with Cluster C1. Finally, Cluster C3 (yellow circles in Figure 8b) included the extensive central-eastern portion of the Sahel, including large territories of Chad, Sudan and Nigeria.

Notable differences and similarities among the three clusters in terms of their response to climatic indices and long-term trends in SPEI-12 were also observed (Figure 8c), highlighting a spatial heterogeneity of drought conditions across the Sahel.

Clusters C1 and C2 exhibit broadly similar patterns, with negative correlations dominating the relationship between SPEI-12 and most climatic indices. Both clusters show particularly strong negative correlations with the GMT and IPWP, suggesting that these indices play a key role in driving aridity in these regions. However, the magnitude of these relationships differs, with C1 generally showing slightly stronger negative correlations than C2. A key distinction is that C2 demonstrates a stronger positive correlation of SPEI-12 with Sahel P (0.24), which suggests this cluster benefits from regional precipitation patterns in mitigating drought, unlike C1, where Sahel P shows a weaker positive effect (0.15). Cluster C3 stands out from the other two clusters, showing weaker negative correlations of SPEI-12 with most indices and even positive correlations with some, such as the AMO and AMM. This indicates that the drivers of drought in C3 are less linked to the same global climatic indices that strongly influence C1 and C2. Additionally, the weaker negative correlation with GMT and IPWP in C3 highlights a distinct climatic regime compared to the other clusters.




In terms of long-term drought trends, Cluster C2 shows the most pronounced worsening, with the lowest Z-value (-5.04). This indicates a sharp decline in SPEI-12, reflecting severe drought intensification. Cluster C3, while still experiencing a negative

trend (-2.98), exhibits a less severe decline compared to C2, with its moderate correlations suggesting more stable conditions overall. Cluster C1 occupies an intermediate position, with a relatively mild decreasing trend (-0.93) and weaker correlations with regional precipitation patterns, pointing to a more gradual but persistent worsening of drought conditions.

Overall, while C1 and C2 share similarities in their drought responses, C2 is more vulnerable to severe drought intensification, whereas C3 differs fundamentally, with weaker correlations and a slower trend toward worsening conditions. These differences

underline the need for cluster-specific approaches to understanding and addressing drought impacts in the Sahel.

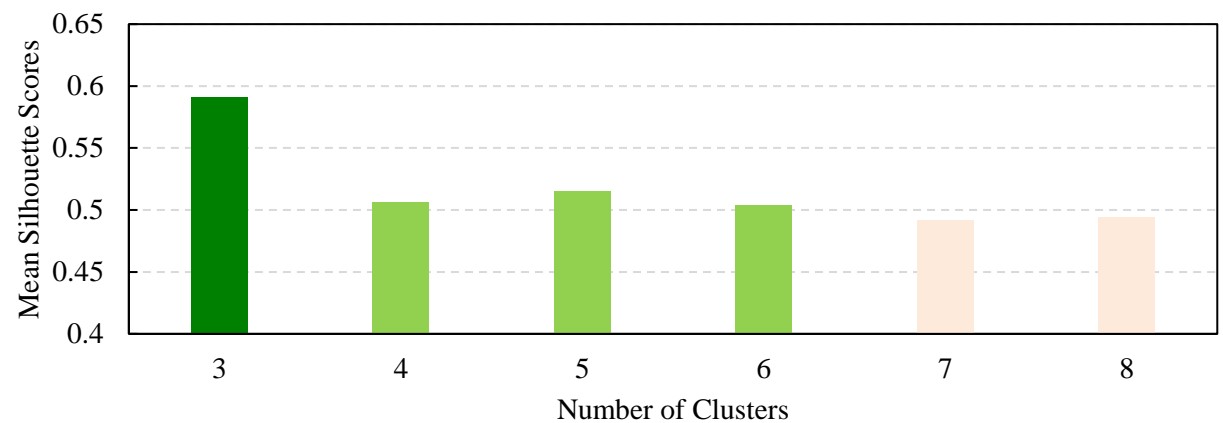

(a)

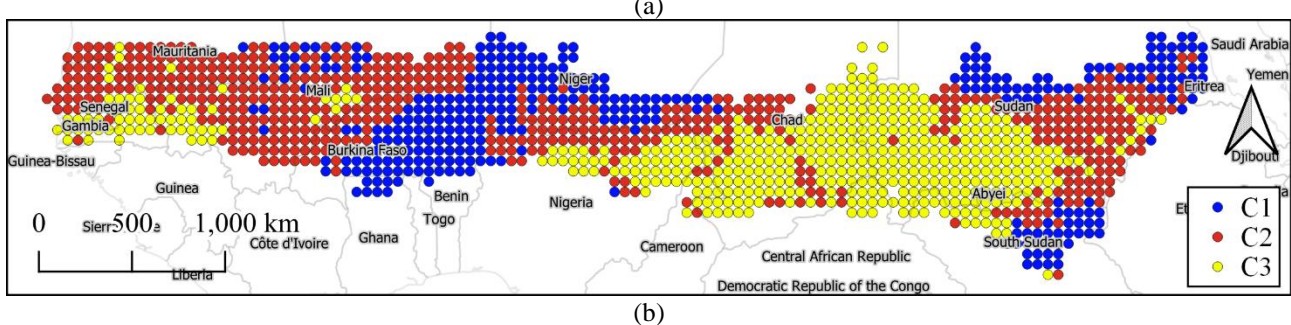

(b)





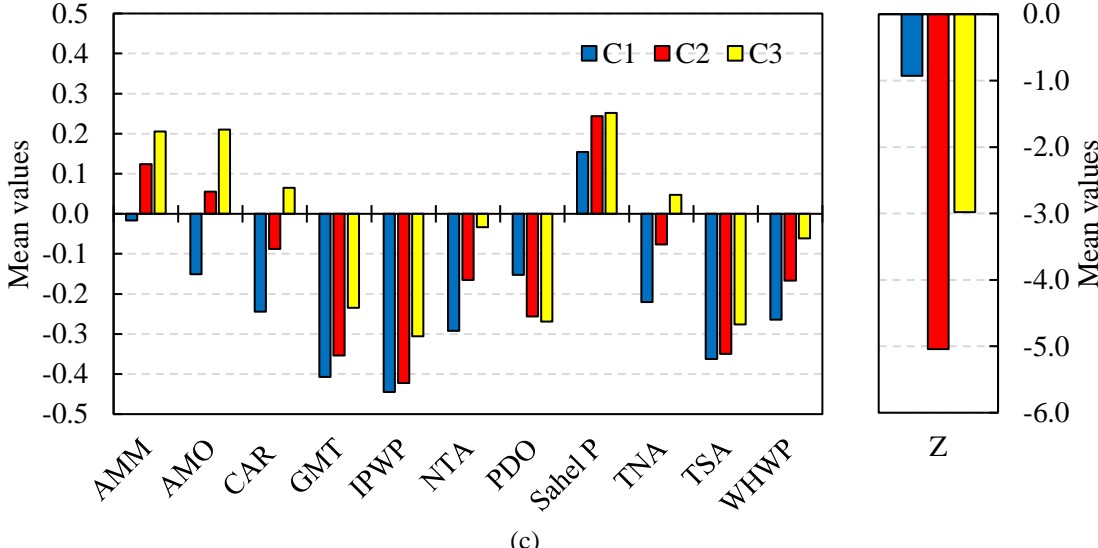

(c)

**Figure 8. Clustering analysis: mean Silhouette Scores calculated for cluster numbers ranging from 3 to 8 (a); K-means clustering map of the Sahel Region (b); mean values of correlations between SPEI-12 and selected climatic indices and the Z statistic from the SK test performed on SPEI-12 for each cluster (c).**

The impact of the correlation between each climatic index and the SPEI-12 on the clustering process was assessed by combining the RF model with SHAP analysis. RF was employed to classify data points into clusters, enabling SHAP to quantify the influence of each climatic variable on the clustering process (see the Beeswarm plots in Figure 9).

For Cluster C1, the correlation between AMO and SPEI-12 had the strongest influence on clustering, followed by the CAR and TNA indices (SHAP values ranging from -0.1 to 0.3). In contrast, correlations related to GMT, IPWP, and TSA exhibited the lowest impact, with SHAP values between -0.05 and 0.05. The distribution of SHAP values for the correlations of most climatic indices was notably skewed toward negative values, meaning their influence on the clustering process was generally negative. Higher values of these correlations (highlighted by red circles in the Beeswarm plots) were predominantly linked to negative SHAP values, except for the PDO index. This suggests that increases in these correlations with climatic indices tend to negatively affect how data points are grouped into clusters. Additionally, the skewed distribution indicates that negative SHAP values were more frequent, reinforcing the overall negative influence of the correlations of these indices with SPEI-12 on the clustering process.

For Cluster C2, the correlation between AMO and SPEI-12 exhibited the highest impact on the clustering (like C1), followed by the NTA and the CAR indices (SHAP values ranging from -0.25 to 0.20). In contrast, correlations with Sahel P, PDO, and TSA demonstrated the lowest impact (SHAP values ranging from -0.10 to 0.05). Unlike C1, in C2, the distribution of SHAP values was not skewed towards negative values. Instead, some correlations exhibited sample concentrations of both negative and positive SHAP values. In contrast to C1, where the highest correlations of each climatic index were primarily associated with negative SHAP values, C2 showed no clear relationship between the magnitude of an index correlation and its impact on



the clustering. Therefore, the high or low values of an index in C2 did not consistently correspond to strong positive or negative SHAP values.

For Cluster C3, the correlation between NTA and SPEI-12 had the strongest influence on clustering, followed by the AMO and WHWP indices (SHAP values ranging from -0.20 to 0.25). In contrast, correlations with TSA, Sahel P, and PDO had the least impact, with SHAP values between -0.1 and 0.1. Like Cluster C1, the distribution of SHAP values in C3 was skewed

toward negative values, indicating that correlations with most climatic indices exerted a negative influence. However, key differences emerged. Notably, in C3, the highest values of each correlation were generally associated with positive SHAP values, whereas in C1, they were predominantly linked to negative values. This suggests that while both clusters are influenced by climatic factors, their responses differ. In both cases, most correlations with climatic indices exert a negative impact on clustering. However, in C3, when a climatic index reaches its highest value, it often contributes positively to the clustering

process, making high-value data points more likely to be part of C3. In contrast, in C1, high correlations of the same indices are less common, indicating a predominantly negative influence on cluster formation.



**Figure 9. SHAP Beeswarm plot for the three clusters.**



## 4 Discussion

### 4.1 Reconceptualizing Drought Dynamics in the Sahel: A Multiscale Perspective

The findings of this study reveal a profound reconfiguration of hydroclimatic regimes across the Sahel, with an alarming 57.5% of grid cells exhibiting a statistically significant decline in SPEI-12. This extensive drying is not a localized phenomenon but rather a manifestation of large-scale climatic reorganization, where the intensification of drought aligns with a convergence of atmospheric and oceanic anomalies. The most affected regions—the western Sahel (Senegal, Gambia, Mali), southeastern Sahel (South Sudan), and northern-central Sahel (Chad)—serve as hydroclimatic sentinels, encapsulating the broader destabilization of the West African monsoon system. Yet, amidst this widespread aridification, a contrasting signal emerges between Burkina Faso and Nigeria, where a tendency toward wetter conditions underscores the region's inherent climatic heterogeneity and the nonlinear interplay of local and global forcing mechanisms.

The intensification of drought correlates with unequivocal global warming signatures, as reflected in the strong increasing trends of GMT (Z = 28.70) and IPWP (Z = 27.83). These indices do not merely co-evolve with regional drying; rather, they act as thermal amplifiers, accelerating evapotranspiration rates, altering atmospheric moisture gradients, and modulating land-atmosphere feedback in ways that redefine conventional paradigms of drought causality. Meanwhile, the observed decline in TNI, Solar Flux, and Sahel P suggests a shift in moisture transport dynamics and radiative forcing, further reinforcing the complexity of hydrological reorganization in the region.

Beyond linear associations, the correlation structure between global climate drivers and regional drought variability reveals an intricately woven network of teleconnections that challenge traditional dichotomies of cause and effect. The Atlantic Multidecadal Oscillation (AMO) emerges as a bifurcated influence, exerting positive correlations in western and central-eastern Sahel but negative correlations in central-western regions such as Burkina Faso. This spatially divergent response suggests that the AMO does not exert uniform control over Sahelian drought but rather interacts with localized boundary conditions in ways that defy simplistic interpretations. Similarly, GMT and IPWP exhibit strong negative correlations with SPEI-12 (-0.76 and -0.71), reinforcing their role as primary drought intensifiers, while Sahel P maintains a positive correlation (0.22), acting as a partial counterbalance to the prevailing drying trend. The weaker and inconsistent influence of AO and NAO underscores the selective and spatially constrained nature of extratropical climatic influences on the Sahel.

The application of K-means clustering transcends conventional regional classifications, revealing three distinct drought-prone domains that reflect not only geographic coherence but also fundamentally different climate-drought interaction mechanisms. Cluster C1 (central-western Sahel, primarily Niger) is predominantly governed by global warming indices, suggesting that anthropogenic climate change plays an outsized role in dictating its hydroclimatic trajectory. Cluster C2 (western Sahel, including Senegal, Mauritania, and Mali) exhibits the most severe drought intensification (Z = -5.04), positioning it as a critical hotspot for future hydroclimatic vulnerability. In contrast, Cluster C3 (central-eastern Sahel, including Chad and Sudan) demonstrates weaker correlations with global indices, indicating that regional-scale processes and localized land-atmosphere interactions may exert a more dominant control over its drought evolution.





Crucially, the SHAP-driven clustering approach provides a paradigm shift in the conceptualization of drought regimes, moving beyond conventional classifications to quantify the relative influence of climate drivers with an unprecedented level of
interpretability. The identification of AMO and NTA as primary clustering drivers, alongside the dominant role of GMT and IPWP in drought intensification, signals a fundamental restructuring of the region's hydroclimatic identity. This data-driven revelation challenges deterministic models of Sahelian drought and underscores the need for a more nuanced, machine-learning-informed understanding of climate variability and its cascading hydrological implications.

### 4.2 A Reassessment of Climate-Drought Interactions in Light of Existing Literature

A comparative analysis with previous studies on the relationship between drought and climatic indices provides a critical context for interpreting the present findings. The work of Okonkwo (2014), which explores precipitation variability in the Sahel in relation to climate indices, aligns with the present study by confirming the strong influence of the Atlantic Multidecadal Oscillation (AMO). Okonkwo (2014) demonstrates that the warm and cold phases of the AMO are associated with increased and decreased precipitation, respectively—a pattern that has been reaffirmed here. However, while previous
studies have largely treated AMO as a broad-scale modulator of precipitation, the SHAP-driven clustering approach applied in this study reveals that AMO's influence is spatially heterogeneous, exhibiting positive correlations with SPEI-12 across much of the Sahel but negative correlations in specific subregions, particularly Burkina Faso and Niger (Cluster C1). This nuanced perspective challenges the assumption of uniform AMO control over Sahelian hydroclimatic variability and underscores the need for regionally adaptive models.

Similarly, Ndehedehe et al. (2020) investigate the correlation between climatic indices and drought variability using both SPEI and SPI, confirming that AMO exerts a dominant influence over precipitation patterns in the central Sahel. Their findings resonate with the present study, reinforcing AMO's multi-scalar control over drought evolution. However, while Ndehedehe et al. (2020) highlight a significant role of the Pacific Decadal Oscillation (PDO) in modulating Sahelian drought conditions—reporting a negative correlation ($r = -0.53$) between PDO and SPI—the present analysis, based on SPEI-12, identifies a weaker
but still notable negative correlation (up to -0.40) in the western and central-eastern Sahel. This result is corroborated by Lüdecke et al. (2021), who also identify a negative PDO-rainfall relationship. These findings reinforce the multidecadal Pacific-Sahel teleconnection but suggest that its strength and consistency depend on the drought metric employed and the temporal scale of analysis.

    The relationship between North Tropical Atlantic (NTA) Sea Surface Temperatures (SSTs) and Sahelian rainfall, explored by
Wane et al. (2023), further illustrates the intricacies of ocean-atmosphere interactions. Their findings indicate that positive SST anomalies in the NTA enhance rainfall in the western Sahel, while negative anomalies suppress it. This process is driven by changes in atmospheric circulation and moisture transport. The present study confirms the remarkable role of NTA in shaping drought dynamics and clustering patterns. However, an inverse correlation between NTA and SPEI-12 is observed, deviating from the rainfall-based perspective provided by Wane et al. (2023). This divergence is likely attributable to the





fundamental difference between precipitation anomalies and drought indices incorporating evapotranspiration, highlighting the necessity of adopting integrated hydroclimatic indicators when assessing drought mechanisms.

The complex interaction between hydrological drought variability and large-scale climatic drivers is further explored by Ogunrinde et al. (2024), whose study on SPEI-based drought analysis in Nigeria identifies an increasing frequency of drought events, consistent with the present findings. Their correlation analysis ranks the influence of climate indices on drought as SOI

> NAO > AMO, whereas in this study, NAO and SOI exhibit only weak correlations with SPEI and negligible impacts on clustering outcomes. A notable discrepancy emerges in the regional breakdown of drought influences: while both studies identify AMO as a key determinant of Nigerian drought conditions, the relationships with NAO and SOI diverge considerably. In contrast to Ogunrinde et al. (2024), where NAO and SOI exert a measurable impact, the present study finds their effect to be statistically insignificant. These differences likely stem from methodological variations, including the spatial domain of

analysis, the clustering methodology employed, and the timescales considered. More broadly, these results underscore the complexity of climate-drought interactions in the Sahel, suggesting that the influence of extratropical climate oscillations is neither spatially uniform nor temporally consistent, but instead varies according to local climatic regimes and feedback processes.

The present study builds upon and extends prior research by moving beyond conventional correlation analyses, incorporating

a multi-method approach that disentangles the mechanistic drivers of drought at an unprecedented level of granularity. Previous studies have largely treated climatic indices as static modulators of Sahelian precipitation, relying on broad correlation patterns to infer causality. However, by integrating SHAP-driven clustering, this study advances the conceptual understanding of drought variability by quantifying the individual contributions of each climate driver, demonstrating that their influence is often localized, nonlinear, and scale-dependent.

This refined perspective challenges the notion of uniform climate-drought relationships, advocating for a more adaptive, ML-informed approach to hydroclimatic research. The findings emphasize the importance of re-evaluating deterministic frameworks in favor of probabilistic, interpretable, and data-driven methodologies that better capture the dynamic nature of Sahelian drought evolution.

### 4.3 The Transformative Role of SHAP in Clustering Analysis

The integration of SHAP into the clustering process represents a paradigm shift in the interpretation of spatial drought patterns, offering a level of transparency and insight that traditional clustering methods cannot achieve. Unlike conventional approaches, which rely on distance metrics and variance minimization, SHAP quantifies the contribution of each climatic index to the formation of clusters, thus unveiling the underlying drivers of drought variability across the Sahel. By coupling SHAP with ML models such as RF, the clustering process transitions from a purely statistical exercise to an interpretable and

mechanistically insightful analysis of climate-drought interactions.

Traditional clustering methods, such as K-means or Hierarchical clustering, provide spatial groupings of homogeneous regions based on shared climatic characteristics. However, these methods inherently lack explanatory power, as they do not clarify





which variables are most influential in shaping the clustering outcomes. This limitation is particularly problematic in contexts such as the Sahel, where climatic indices are often highly correlated, and their combined influence on drought patterns may

remain obscured. The implementation of SHAP addresses this gap by providing an additive and interpretable measure of variable importance, allowing for a deeper understanding of the mechanistic relationships between climatic drivers and cluster formation.

In this study, the application of SHAP revealed distinct patterns in the influence of climatic indices on clustering outcomes, highlighting the nuanced interplay between oceanic oscillations and regional drought variability. For example, indices such as

the Atlantic Multidecadal Oscillation (AMO) and the North Tropical Atlantic (NTA) Sea Surface Temperature (SST) emerged as key drivers of cluster differentiation. While the AMO generally exhibited a positive correlation with SPEI-12 across much of the Sahel, regions such as Burkina Faso and Niger displayed divergent patterns, where the AMO was negatively correlated with SPEI-12. Conversely, the NTA demonstrated a consistently negative correlation, indicating that decreases in NTA SSTs correspond to increases in SPEI-12 and vice versa. These findings underscore the spatial heterogeneity of climatic responses

to oceanic oscillations, suggesting that local hydroclimatic dynamics mediate the influence of large-scale drivers in ways that differ across the Sahel.

A direct comparison of clustering results with and without SHAP further underscores its added value. Without SHAP, clustering results provide spatial groupings based on statistical similarity but offer no insight into the causal mechanisms underlying these patterns. By incorporating SHAP, however, each cluster is characterized not only by its geographic and

climatic composition but also by a quantitative attribution of the contribution of each climatic index to its formation. This approach transforms the clustering process into a comprehensive framework for understanding regional drought dynamics, enabling the identification of the most influential climatic drivers and their potential causal links to local hydrological responses.

The practical implications of SHAP-driven clustering extend beyond methodological innovation. By pinpointing the dominant

climatic indices shaping regional drought patterns, this approach equips policymakers and resource managers with actionable insights for targeted intervention. For instance, in regions where the AMO exerts the strongest influence, monitoring and mitigation efforts can be tailored to anticipate its phase shifts and associated hydroclimatic impacts. Similarly, the strong negative correlation between NTA and SPEI-12 highlights the necessity of prioritizing ocean-atmosphere interaction monitoring to enhance early warning systems. In a region like the Sahel, where climate variability is shaped by multiple

interacting factors, such interpretability is critical for effective drought mitigation and adaptation strategies.

In summary, the integration of SHAP into clustering analysis redefines the analytical process, bridging the gap between data-driven classification and physical climate dynamics. This methodology not only enhances the reliability and interpretability of clustering results but also provides a nuanced and actionable understanding of the factors driving drought variability in the Sahel. The findings underscore the necessity of moving beyond traditional clustering techniques, advocating for the adoption

of explainable methodologies that can better inform science-based climate adaptation strategies.





### 4.4 Limitations and Future Directions

While this study provides a comprehensive and data-driven framework for assessing drought variability in the Sahel, certain limitations warrant consideration. One inherent constraint lies in the geographical scope of the analysis. Although the Sahel

represents a vast and climatically significant region, the findings are inherently tailored to its semi-arid hydroclimatic regime. As a result, the generalizability of the proposed methodology to regions with markedly different climatic and hydrogeological conditions remains an open question. Future research should seek to extend this framework to diverse environmental contexts, including humid tropical zones, arid desert landscapes, and temperate regions, to evaluate the transferability and adaptability of the approach across varying hydroclimatic gradients. Such an extension would provide a more holistic understanding of

drought dynamics, revealing how different climatic drivers modulate hydrological extremes under contrasting environmental forcings.

Additionally, while this study successfully integrates trend analysis and explainable clustering, further investigation is needed to explore its performance in highly anthropized environments, where urbanization, land-use change, and water extraction exert non-climatic controls on drought evolution. Expanding the application of this methodology to regions experiencing rapid

demographic expansion and infrastructural development—such as peri-urban areas with increasing groundwater dependency—would offer crucial insights into how human-induced modifications interact with large-scale climate drivers to shape drought risk. Likewise, applying this approach to colder climates would enable an assessment of its robustness in regions where snowpack dynamics, freeze-thaw cycles, and permafrost degradation introduce additional complexities to hydrological variability.

Beyond spatial expansion, methodological advancements represent a key avenue for future research. While the integration of SHAP-driven clustering with traditional statistical techniques has enhanced the interpretability of drought patterns, the incorporation of hybrid ML models and advanced change-point detection algorithms could further refine the detection of non-stationary behaviors in drought variability. ML models—particularly deep learning architectures and ensemble learning frameworks—hold promise for capturing high-dimensional dependencies within climate-drought interactions, thereby

improving predictive accuracy. Similarly, the application of advanced change-point detection methods could enhance the ability to identify abrupt hydroclimatic regime shifts, offering a more granular perspective on the evolving nature of drought risk.

Future studies should also consider integrating multi-source datasets, including remote sensing observations, high-resolution reanalysis products, and socio-economic indicators, to develop a more holistic and cross-disciplinary framework for drought

assessment. This would facilitate a transition from a purely climatological perspective to a socio-hydrological paradigm, acknowledging the feedback between climate variability, human adaptation strategies, and water resource sustainability.

Ultimately, the continued evolution of interpretable and adaptive methodologies is essential to advancing the understanding of drought risk in a rapidly changing global climate. By bridging the gap between statistical inference, ML, and hydroclimatic





process understanding, future research has the potential to redefine drought analysis, enabling more effective climate
adaptation and water resource management strategies at both regional and global scales.

## 5. Conclusion

This study presents a comprehensive framework for assessing drought variability in the Sahel by integrating trend analysis, cross-correlation, and an innovative SHAP-driven clustering approach. The analysis revealed that 57.5% of the region exhibits a significant drying trend in SPEI-12, particularly in the western and southeastern Sahel, driven by increasing temperatures
and declining precipitation. Conversely, 19.3% of the region shows statistically significant wetting trends, highlighting the spatial heterogeneity of drought evolution primarily through increased evapotranspiration and reduced soil moisture availability. At a regional scale, AMO and NTA emerged as key modulators of drought variability, influencing distinct drought-prone zones. The application of SHAP-driven clustering provided a more interpretable and data-driven classification of drought risk areas, distinguishing three distinct clusters with varying hydroclimatic influences.

Beyond its methodological contributions, this study underscores the importance of region-specific drought adaptation strategies to enhance water resource resilience in the Sahel. The findings emphasize the necessity for data-informed decision-making, leveraging AI-enhanced drought assessment tools to improve regional water management strategies. Future research should focus on integrating predictive modeling frameworks and expanding this methodology to other drought-prone regions worldwide, fostering a more comprehensive understanding of climate-driven hydrological extremes.

By bridging advanced statistical analysis with explainable AI techniques, this study provides a significant step forward in interpreting climate impacts on regional water security, offering valuable insights for policymakers and hydrological researchers alike.

**Data availability statement**

The SPEI gridded data from the Global SPEI Database (GSD) were available at the following website: https://spei.csic.es/

**Author contribution**

Fabio Di Nunno contributed with conceptualization, data curation, formal analysis, investigation, methodology, software, visualization, writing – original draft. Yıldız Mehmet Berkant contributed with data curation, investigation, formal analysis, software, visualization, writing – original draft. Francesco Granata contributed with supervision, conceptualization, data curation, formal analysis, investigation, methodology, software, validation, visualization, writing – original draft.



## Competing interests

The authors declare that they have no conflict of interest.

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
