# Peer review of "Decoding the Architecture of Drought: SHAP-Enhanced Insights into the Climate Forces Reshaping the Sahel"

_EGUsphere, 2025_

## Author Comment (AC1)

**Reviewer 1**

This paper investigates how various climate indices impacts drought assessment measured by SPEI, based on an explainable-AI framework. Below are my major concerns followed by minor comments.

We would like to thank the Reviewer for the careful review of our manuscript. We have revised the manuscript, taking into consideration all the comments. During the revision, we also made numerous changes. All these changes have significantly improved the quality and presentation of the manuscript, and we hope the current version is acceptable for publication. Below, we provide our point-by-point responses to the specific review comments.

**Major comments:**

1.  While the authors claim that they use an explainable-AI framework, the methods section has limited details about SHAP and how the AI is explainable. Random Forest, while tree-based, does not embed physical mechanisms as a priori. Relevant explanations in the manuscript are also very brief. For example, there are no details about how the feature value and SHAP value work, and what information the beeswarm plot conveys in Fig. 9. The caption of Fig. 9 is also very short. Line 345 "increases in correlations with climatic indices tend to negatively affect how data points are grouped into clusters" also requires a clear physical interpretation.

    Thanks for the constructive comments regarding the explainability of our AI framework and the presentation of SHAP results. In the revised version of the manuscript, we have substantially expanded the Methods section to clarify how SHAP values function within our Random Forest classifier to provide interpretability. Specifically, we now explain that SHAP values quantify the marginal contribution of each climatic index correlation to the cluster assignment, with positive and negative values indicating features that respectively increase or decrease the likelihood of cluster membership.

    Furthermore, the caption and main text describing Figure 9 have been enhanced to clearly interpret the beeswarm plot: the x-axis reflects the SHAP value impact on clustering, while the color gradient encodes the actual feature (climatic index correlation) value, allowing a nuanced understanding of how feature magnitude and direction affect classification. We also provide detailed cluster-wise analyses of feature importance distributions and their directional effects, clarifying the roles of dominant versus minor indices.

    Regarding the statement "increases in correlations with climatic indices tend to negatively affect how data points are grouped into clusters," we have widely revised the text to offer a clearer physical interpretation within the context of clustering behavior, emphasizing how changes in feature correlations influence the clustering structure through their SHAP contributions.

We trust these clarifications address your concerns and improve the transparency and interpretability of our explainable-AI approach. Below the revised text for the SHAP analysis:

*To evaluate the relative influence of each climatic index on the clustering process and assess the predictive performance of the classifier, we employed an explainable AI approach that integrates a Random Forest (RF) classifier with SHAP. The RF model, a robust tree-based ensemble algorithm, effectively captures complex nonlinear interactions among variables but lacks inherent interpretability. To address both model performance and transparency, a comprehensive protocol was implemented.*

*First, the dataset was split using stratified sampling into training (90%) and testing (10%) subsets to preserve the original class distribution. A Random Forest classifier (100 estimators, criterion=Gini, random_state=42) was trained on the training data, and standard evaluation metrics—accuracy, class-wise precision, recall, F1-score, and the confusion matrix—were computed on the test set. The model achieved an accuracy of 0.985 on the independent test set. Class-wise precision, recall, and F1-scores were all above 0.97, confirming the classifier's strong discriminative power (see Table S4). Second, model explainability was addressed using SHAP values computed through the TreeExplainer framework. Beeswarm plots were generated for each cluster to visualize the magnitude and direction of feature contributions. Moreover, for each cluster, mean absolute SHAP values were computed for each feature, and a bootstrap procedure (n = 100) was performed to calculate 95% confidence intervals, providing statistical robustness to the importance rankings.*

*SHAP values represent the marginal impact of each feature on a model's prediction, averaged over all possible feature subsets. In this context, a positive SHAP value indicates that the feature increases the likelihood of a data point being assigned to a particular cluster, while a negative value suggests a suppressing effect. In the SHAP beeswarm plots (Figure 9), the x-axis represents SHAP values—the impact of each feature on the clustering outcome—while the color gradient (Feature value) encodes the actual correlation value between the climatic index and SPEI-12 for each data point, ranging from low (blue) to high (red). This dual encoding enables a nuanced interpretation of the model's behavior: the position along the x-axis reflects the strength and direction of influence, while the color reveals whether strong or weak correlations drive the effect.*

*The SHAP beeswarm plots for Clusters C1, C2, and C3 provide a comprehensive breakdown of the influence that each climate index exerts on the Random Forest classifier's clustering outcomes. Each plot reveals both the magnitude and direction of influence through SHAP values, offering insight into the discriminative role of individual features in defining cluster membership.*

*In Cluster C1, the AMO, CAR and TNA emerged as the most influential variables, with mean absolute SHAP values of 0.088, 0.72 and 0.059, respectively (see Table S5). Their distributions are notably skewed toward positive SHAP*

values, with dense concentrations between 0.05 and 0.15. This pattern indicates a strong and consistent association between high index values and increased likelihood of C1 classification. Moderately influential indices such as WHWP, NTA, and AMM present narrower spreads (-0.05 to 0.2) and more symmetric profiles, suggesting subtler but still directional contributions. Conversely, indices like PDO, GMT, IPWP, and TSA show very limited SHAP influence, with values clustered near zero and minimal dispersion, highlighting their negligible role in defining this cluster.

Cluster C2, in contrast, was characterized by AMO and the NTA indices as the most important features (mean absolute SHAP equal to 0.096 and 0.084, respectively), followed by CAR and TNA. These variables show significant spread on both sides of zero, implying a bidirectional influence where both high and low values can affect classification, depending on the context. Secondary contributors such as IPWP, AMM, and GMT exhibit tighter distributions centered around zero but with occasional asymmetries, pointing to context-dependent roles. Sahel P, PDO, and TSA remain minimally influential, with narrow SHAP ranges and modes at or near zero. Compared to Cluster C1, the SHAP profiles in C2 suggest greater interaction complexity among variables rather than dominance by a few.

For Cluster C3, NTA dominated the feature importance ranking (mean absolute SHAP: 0.102), followed by AMO (0.080) and WHWP (0.072). These distributions are distinctly positively skewed, and the color gradient confirms that high feature values strongly align with positive SHAP contributions. Variables such as CAR, TNA, and IPWP follow a similar, though slightly less pronounced, pattern. Mid-tier contributors like GMT, AMM, and TSA are more symmetrically distributed, with modal SHAP values just above zero. Finally, Sahel P and PDO again register as the least impactful, mirroring the behavior observed in the other clusters.

Across all three clusters, a consistent pattern emerges in the relative importance of certain indices. AMO, and NTA are among the most influential features throughout, though the nature of their impact differs. In Clusters C1 and C3, their SHAP distributions are positively skewed, indicating a clear, directional relationship between high index values and cluster membership. In contrast, Cluster C2 exhibits more symmetric SHAP profiles, highlighting bidirectional effects and greater context dependency.

Another key distinction lies in the degree of feature dominance. Cluster C1 and C3 are shaped by a small subset of highly influential variables with strong directional effects, whereas Cluster C2 displays a more distributed influence among multiple variables with less sharply skewed contributions.

Low-impact indices such as PDO and TSA consistently show minimal influence across all clusters. Their SHAP values remain centered around zero with low density, suggesting that these variables have limited utility in discriminating among the regimes captured by the clustering model.

2. The paper claims a methodological advancement, but the literature review gives limited coverage of studies that use conventional approach. The Discussion should (a) compare the present results with key earlier studies that relied on traditional methods, and (b) explain why the proposed framework leads to superior or complementary results.

We appreciate the Reviewer's insightful comment. In the revised manuscript, we have substantially expanded Section 4.3 (now titled *"Advancing Hydrological Clustering: From Conventional Methods to SHAP-Enhanced Insights"*) to address both parts of the suggestion:

**(a) Comparison with key earlier studies using traditional clustering methods:** We now discuss several representative studies that applied conventional clustering techniques in hydrological and drought-related research. These include K-means applications to global PDSI patterns (Najafi and Khanbilvardi, 2018), hierarchical and fuzzy clustering in South Korea (Azam et al., 2018) and western India (Goyal and Sharma, 2016), and a recent combination of clustering and forecasting in Southern Italy (Di Nunno and Granata, 2023). These studies are valuable in capturing statistical similarities in drought behavior and in supporting regionalization. However, they typically rely on distance-based metrics and offer limited ability to explain *why* clusters form or *which* variables most influence regional drought regimes.

**(b) Explanation of the proposed framework's superiority or complementarity:** We clarify how our framework advances these conventional methods by incorporating SHAP (SHapley Additive exPlanations) into the clustering process. This innovation transforms clustering from a statistical to a mechanistically interpretable task. It enables us to quantify the influence of each climatic driver on each cluster and to identify both dominant and negligible contributors to drought variability—capabilities that are absent in traditional clustering. By doing so, our approach supports more transparent attribution, improves decision-making for early warning and adaptation strategies, and complements existing clustering work by embedding causal interpretability into spatial drought analysis.

We believe this addition strengthens the Discussion and clearly supports the claim of methodological advancement.

3. Abstract line 10, it is inappropriate to state that XX% has stat sig trend because there could be spatial autocorrelation that inflate counts of significance. Same thing for Line 201-202, Line 372, Line 538. A relevant paper is Wilks, D. S. "On "field significance" and the false discovery rate." *Journal of applied meteorology and climatology* 9 (2006): 1181-1189.

We sincerely thank the Reviewer for this important observation regarding spatial autocorrelation and its implications for interpreting statistical significance in gridded climate data. We fully agree that spatial dependence can lead to an inflated number of statistically significant tests, as outlined in Wilks (2006), and we are aware of the limitations this presents in field-scale trend analysis.

In our manuscript, the reported percentages of statistically significant trends are intended to offer a descriptive overview of the spatial extent of the observed patterns, rather than to suggest a rigorous count of independent significant results. These trends are further interpreted within the broader context of spatial coherence, regional climatic patterns, and clustering analysis. Moreover, the robustness of our findings is supported through multiple complementary techniques (e.g., cross-correlation and SHAP-driven clustering), which together provide a multi-dimensional view of drought evolution in the Sahel.

We have chosen to retain the current text for clarity and interpretability but are willing to insert a section in the discussion acknowledging the potential effect of spatial autocorrelation and citing Wilks (2006), should the reviewer or editor deem it necessary:

*Moreover, while the percentage of grid cells showing statistically significant trends is reported to convey a general sense of spatial extent, we acknowledge that such figures can be affected by spatial autocorrelation, potentially inflating the number of significant results. As such, these values should be interpreted cautiously, with emphasis placed on coherent spatial patterns rather than individual significance. This limitation, discussed in the literature (e.g., Wilks, 2006), highlights the importance of adopting field significance approaches in future work to address spatial dependencies in gridded climate data.*

We trust that the holistic and multi-methodological framework adopted in our analysis mitigates the risk of overinterpretation of localized significance and maintains the scientific validity of the conclusions presented.

4. It is unclear what "climate indices" means. Broadly speaking, SPEI itself can also be a climate index. The authors should highlight large-scale climate variability or provide a formal definition of climate indices.

We thank the reviewer for this helpful comment. We agree that the term "climate indices" can be ambiguous and that clarification is necessary. In this study, "climate indices" refers specifically to standardized metrics that represent large-scale modes of atmospheric and oceanic variability (e.g., AMO, ENSO, NAO), which influence regional climate patterns. To address this, we have added a formal definition in Section 2.1 of the manuscript:

*Drought assessment in the Sahel is complicated by the complex, nonlinear, and dynamic nature of atmospheric processes, which challenge the accurate representation of spatial–temporal patterns, multi-scale interactions, and the influence of extreme events and topographic variability. To address these complexities, this study incorporates time series of various climate indices into the modeling framework.*

*In this context, climate indices refer to large-scale indicators of atmospheric and oceanic variability derived from standardized measurements such as sea surface temperature (SST), sea-level pressure, and wind anomalies over specific regions. Examples include the AMO, GMT and North Atlantic Oscillation (NAO). Unlike drought indicators*

*such as SPEI, which quantify regional hydroclimatic conditions, climate indices capture broader patterns of variability that serve as external drivers of local drought dynamics.*

*These indices offer critical insights into the mechanisms regulating regional drought variability. For instance, warm phases of the AMO are associated with increased rainfall in the Sahel, whereas El Niño events often lead to drier conditions (Okonkwo, 2014). The 12-month SPEI timescale was selected to reflect both seasonal and interannual climate variability, enabling the detection of annual hydrological responses to the prevailing phases of large-scale climate drivers. While some indices, such as the AMO, operate on multidecadal timescales, their current phase can still exert influence on precipitation patterns within a given year. Thus, the 12-month period is not intended to resolve long-term climate variability itself, but rather to integrate its effects as expressed in a single year's climate system. This timescale effectively captures the cumulative influence of slow-acting processes such as oceanic and atmospheric anomalies, allowing SPEI to reflect integrated climate impacts on precipitation and evapotranspiration. As a result, the use of climate indices alongside long-term SPEI enhances the ability to identify meaningful correlations, detect persistent drought trends, and better understand the climatic forces shaping drought conditions in the region.*

5.  Table 1 lists many indices, but the manuscript does not explain why each is relevant to Sahel/African hydroclimate. Please justify the inclusion of each index or focus on a subset with documented influence on the region, similar to the description of "Sahel Precipitation".

    The indices listed in Table 1 were selected for their potential influence on atmospheric and oceanic conditions that directly or indirectly affect the hydroclimate of the Sahel and the broader African region. Table 1 has been updated to include descriptions of each index, highlighting their potential impacts on Africa and the Sahel where relevant, and the data source.

6.  Using two particular cells in Fig. 6 and Fig. 7 is not representative. The two cell is just two out of 1335 SPEI gridded data points in the study region, and there is not a clear rationale for focusing on these cells. It is hard to follow the motivation of the analysis. While the cell in Fig. 6 has the strongest positive correlation between AMO and SPEI, the overall correlation mean is only "modest" at 0.06 (Line 239). How could it support the statement in Line 270, "AMO are closely tied to sub-regional drought dynamics"?

    We thank the Reviewer for the insightful observation regarding the use of specific grid cells in Figures 6 and 7. The intention behind highlighting Cells 2042 and 2319 was not to generalize their behavior to the entire Sahel region, but

rather to provide illustrative examples that represent spatial extremes in the correlation distribution—i.e., the highest positive and negative correlations observed in the domain for AMO and GMT, respectively.

This approach was chosen to help readers better understand how large-scale climatic indices can exert regionally differentiated influences on drought variability. While the mean correlation values for indices like AMO should be modest at the regional scale, the spatial heterogeneity is substantial, as indicated by the wide range and standard deviation of the correlation coefficients (as reported in Table S3). The selected cells exemplify areas where these influences are more pronounced and thus help to illustrate the sub-regional relevance of these correlations.

7. There are mismatches and typos in the manuscript. I suggest the authors carefully read their manuscript throughout. To name only a few: Line 231 refers to Fig. 5 as a "combined box and violin plot," but Fig. 5 is a map. "ahel" in Table 1 should be "Sahel." Line 230 describes Fig. 4 as "maps of correlations … and the most correlated climatic indices," but Fig. 4 shows bar plots for all indices.

We thank the Reviewer for carefully pointing out the inconsistencies and typographical errors in the manuscript.

We have thoroughly reviewed the entire document to correct such issues. Specifically:

Line 231: The description of Fig. 5 has been corrected. We mistakenly referred to it as a "combined box and violin plot," while it is indeed a map. The caption and in-text reference have been revised accordingly.

Table 1: The term "ahel" has been corrected to "Sahel" to accurately reflect the regional classification.

Line 230: The description of Figures 4 and has been revised:

*Figure 4 reports a combined box and violin plots representation of the correlations for all climatic indices, while Figure 5 provides the maps of the correlations between SPEI-12 gridded data and the most correlated climatic indices.*

In addition to these specific issues, we have performed a thorough proofread of the entire manuscript to correct any other typographical or referencing inconsistencies.

8. The manuscript does not specify the data sources for each climate index in Table 1.

Thanks for the comment. As stated above, Table 1 has been updated to include descriptions of each index, highlighting their potential impacts on Africa and the Sahel where relevant, and the data source.

| Climate index | Abbr. | Definition | Data source |
|---|---|---|---|
| Atlantic Meridional Mode | AMM | The AMM describes north-south SST differences in the tropical Atlantic. Its positive phase shifts rainfall northward, increasing Sahel precipitation and reducing drought risk. The negative phase causes southward rainfall shifts, leading to Sahel drought. AMM also affects Atlantic hurricane activity, influencing regional climate variability. | https://psl.noaa.gov/data/timeseries/month/DS/AMM/ |

| | | | |
|---|---|---|---|
| Atlantic Multidecadal Oscillation | AMO | The Atlantic Multidecadal Oscillation refers to natural variations in North Atlantic Ocean sea surface temperatures that occur over periods of 20 to 40 years. In its positive phase, North Atlantic temperatures are above average, leading to hotter summers along the eastern U.S., increased hurricane activity in the tropical Atlantic, and enhanced rainfall in Africa. In its negative phase, cooler Atlantic temperatures are associated with weaker hurricane activity, drought in Africa's Sahel region, and cooler, wetter summers in Europe. The AMO plays a significant role in shaping global climate systems and regional weather patterns, particularly in the North Atlantic region. | https://www.psl.noaa.gov/data/timeseries/AMO/ |
| Arctic Oscillation | AO | The AO influences atmospheric circulation patterns that can extend to the Sahel region by affecting the strength and position of the African Easterly Jet and mid-latitude weather systems. Its negative phase can weaken the jet stream, altering rainfall patterns in West Africa and contributing to Sahel drought or variability in seasonal precipitation. | https://psl.noaa.gov/data/timeseries/month/DS/AO/ |
| Berkeley Earth Surface Temperature | BEST | The BEST dataset provides global and regional surface temperature trends, including detailed temperature anomalies across Africa. These trends are crucial for understanding how warming influences Sahel hydroclimate, as rising temperatures can exacerbate drought conditions and impact rainfall variability in the region. | https://psl.noaa.gov/data/correlation/censo.data |
| Caribbean Index | CAR | The CAR index captures climate variability in the Caribbean, including SST and atmospheric patterns. It influences Atlantic tropical cyclone activity, which can affect West African monsoon dynamics and Sahel rainfall through atmospheric teleconnections. | https://psl.noaa.gov/data/correlation/CAR_ersst.data |
| Eastern Pacific Oscillation Index | EPO | The EPO describes atmospheric pressure anomalies in the eastern North Pacific. Its phases influence the jet stream and temperature patterns in North America, which can indirectly affect West African climate by modulating large-scale atmospheric circulation and teleconnections linked to Sahel rainfall variability. | https://psl.noaa.gov/data/correlation/epo.data |
| Greenland Blocking Index | GBI | The GBI measures persistent high-pressure systems over Greenland. Its positive phase alters North Atlantic circulation, which can influence the West African monsoon and Sahel rainfall by affecting atmospheric patterns that modulate moisture transport into the region. | https://psl.noaa.gov/data/correlation/gbi.ncep.day |
| Global Mean Temperature | GMT | GMT tracks overall atmospheric and ocean warming or cooling trends. Rising global temperatures influence the Sahel by intensifying droughts, altering rainfall patterns, and impacting regional water resources through shifts in the hydrological cycle. | https://psl.noaa.gov/data/correlation/gmsst.data |
| Indo-Pacific Warm Pool | IPWP | The IPWP, with some of the warmest tropical ocean temperatures, drives global atmospheric circulation, including monsoons. Its warming phase enhances convection and rainfall, indirectly influencing Sahel rainfall through shifts in the Walker circulation and global moisture transport. | https://psl.noaa.gov/data/correlation/pacwarm.data |
| The North Atlantic Oscillation | NAO | The NAO index measures sea-level pressure differences between the Azores High and the Subpolar Low. Its phases modulate the North Atlantic jet stream and storm tracks, affecting heat and moisture transport. These changes influence West African monsoon strength and Sahel precipitation by altering atmospheric circulation patterns over the Atlantic. | https://psl.noaa.gov/data/correlation/nao.data |
| The North Atlantic Oscillation (Jones) | NAO (Jones) | Defined by Jones (1997), this NAO index measures the sea-level pressure difference between the Azores High and Icelandic Low. Its phases influence Atlantic atmospheric circulation patterns that affect West African monsoon dynamics and Sahel rainfall variability. | https://psl.noaa.gov/data/correlation/jonesnao.data |
| Niño-1.2 | - | This index covers sea surface temperatures in the eastern equatorial Pacific (80°W–90°W, 10°S–0°), where El Niño and La Niña events typically originate. In its positive phase (El Niño), warmer waters lead to increased rainfall and floods along South America's northwest coast; in its negative phase (La Niña), cooler waters cause drought and promote cold-water upwelling. While its direct effects are regional, Niño-1+2 influences large-scale atmospheric circulation patterns, which can alter the West African monsoon strength and consequently affect Sahel precipitation variability. | https://www.cpc.ncep.noaa.gov/data/indices/ersst5.nino.mth.91-20.ascii |
| Niño 3 | - | The Niño-3 index tracks sea surface temperatures in the eastern equatorial Pacific (150°W–90°W, 5°S–5°N) to monitor El Niño and La Niña events. During El Niño (positive phase), warmer waters cause increased rainfall in western South America, drought in Asia-Pacific, and reduced rainfall in the | https://www.cpc.ncep.noaa.gov/data/indices/ersst5.nino.mth.91-20.ascii |

| | | Sahel. La Niña (negative phase) brings cooler waters, increased storms in Asia-Pacific, and enhanced rainfall in the Sahel. This index influences global atmospheric circulation and tropical rainfall patterns. | |
|---|---|---|---|
| Niño-3.4 | - | The Niño-3.4 index measures sea surface temperatures in the central equatorial Pacific (120°W–170°W, 5°S–5°N) and is a key indicator of El Niño and La Niña events. During El Niño (positive phase), warmer waters lead to increased rainfall along South America's coast, drought and heatwaves in the Asia-Pacific, and reduced rainfall in the Sahel. La Niña (negative phase) brings cooler waters, heavy rains and floods in Asia-Pacific, drought in South America, and enhanced rainfall in the Sahel. Niño-3.4 strongly influences global atmospheric circulation and tropical weather patterns. | https://www.cpc.ncep.noaa.gov/data/indices/ersst5.nino.mth.91-20.ascii |
| Niño-4 | - | The Niño-4 index measures sea surface temperature variations in the central Pacific (160°E–150°W, 5°S–5°N) during El Niño and La Niña events. In its positive phase (El Niño), warmer waters enhance convection in the tropical western Pacific, influencing atmospheric circulation and monsoon patterns, often linked to reduced rainfall in the Sahel. In the negative phase (La Niña), cooler waters lead to drought in the western Pacific and increased rainfall in the central Pacific, sometimes boosting Sahel precipitation. Niño-4 is key for understanding tropical circulation and regional climate variability. | https://www.cpc.ncep.noaa.gov/data/indices/ersst5.nino.mth.91-20.ascii |
| Northern Oscillation Index | NOI | The NOI measures sea level pressure differences between Tahiti (eastern tropical Pacific) and Darwin (western subtropical Pacific). In its positive phase, high pressure dominates the eastern Pacific and low pressure the western Pacific, causing drought in the east and wetter conditions in the west. The negative phase reverses this pattern. The NOI is essential for understanding tropical climate phenomena such as El Niño-Southern Oscillation (ENSO) impacts. | https://psl.noaa.gov/data/correlation/noi.data |
| North Pasific Index | NP | The NP reflects average sea level pressure over the North Pacific, indicating the strength of the Aleutian Low. In its positive phase, a stronger Aleutian Low brings more rainfall to western North America and cooler eastern Pacific waters. In the negative phase, the system weakens, leading to drier conditions and warmer sea surface temperatures. The NP Index is key to understanding Pacific atmospheric circulation and its effects on North American and, to a lesser extent, African climate patterns | https://psl.noaa.gov/data/correlation/np.data |
| North Tropical Atlantic SST Index | NTA | The NTA index represents SSTs in the North Tropical Atlantic. In its positive phase, warmer SSTs enhance convection and tropical cyclone activity, while influencing rainfall patterns across the Atlantic basin. This phase is often linked to increased precipitation in the Sahel. In the negative phase, cooler SSTs reduce cyclone activity and can lead to drought conditions in West Africa. The NTA index is crucial for understanding Atlantic climate variability and its impacts on regional weather systems. | https://psl.noaa.gov/data/correlation/NTA_ersst.data |
| Oceanic Niño Index | ONI | The ONI measures the three-month running average of sea surface temperature anomalies in the Niño-3.4 region of the central tropical Pacific. It is the primary indicator of El Niño and La Niña events within the ENSO cycle. Positive ONI values (El Niño) are linked to drought in regions like Australia and the Sahel, while negative values (La Niña) can enhance rainfall in these areas. The ONI is essential for monitoring ENSO's global impacts on temperature, rainfall, and atmospheric circulation | https://psl.noaa.gov/data/correlation/oni.data |
| Pasific Decadal Oscillation | PDO | The PDO describes long-term shifts in sea surface temperatures and atmospheric pressure across the North Pacific. In its positive phase, the eastern Pacific cools while the western Pacific warms, bringing wetter conditions to western North America and warmer weather in Alaska. In the negative phase, the pattern reverses, causing drought in western North America and reduced marine productivity. The PDO influences multi-decadal climate variability, affecting agriculture, fisheries, and water resources globally, including rainfall patterns in Africa | https://psl.noaa.gov/data/correlation/pdo.data |
| Pacific Meridional Mode | PMM | The PMM is a climate pattern driven by interactions between sea surface temperatures and surface winds in the tropical Pacific. In its positive phase, warmer waters enhance convection and rainfall across the Pacific, often preconditioning El Niño events. In its negative phase, cooler waters suppress convection, leading to drier tropical conditions. The PMM influences the onset | https://psl.noaa.gov/data/timeseries/month/data/pmm.data |

| | | | |
|---|---|---|---|
| | | of ENSO events and plays a key role in shaping tropical and global climate variability, including rainfall over Africa. | |
| Pacific–North American Pattern | PNA | The PNA pattern is a major mode of atmospheric variability in the Northern Hemisphere, reflecting recurring pressure anomalies over the North Pacific and North America. The PNA index is based on standardized 500 hPa geopotential height anomalies at four specific locations. Its phases correlate with temperature and precipitation anomalies across North America. The PNA influences regional weather by modulating the strength and position of the East Asian jet stream, affecting storm tracks and climate patterns. Through atmospheric teleconnections, the PNA can also impact tropical circulations, with potential links to rainfall variability in the Sahel and parts of northern Africa. | https://psl.noaa.gov/data/correlation/pna.data |
| Quasi-Biennial Oscillation | QBO | The QBO is a regular oscillation of easterly and westerly winds in the tropical stratosphere, with a cycle of about 28–30 months. In its westerly phase, tropical cyclone activity increases, especially in the Pacific and Atlantic. In the easterly phase, cyclone formation weakens, and stratospheric ozone distribution shifts. The QBO modulates stratosphere–troposphere interactions and can influence tropical convection, potentially affecting rainfall variability in regions such as the Sahel and equatorial Africa. | https://psl.noaa.gov/data/correlation/qbo.data |
| Sahel Precipitation | Sahel P | Sahel Precipitation refers to the annual rainfall in Africa's Sahel region (south of the Sahara Desert) and is influenced by tropical Atlantic Sea surface temperatures and atmospheric circulation. Positive phases (increased rainfall) improve agriculture and water resources, while negative phases (drought) lead to famine and heightened socio-economic impacts. | https://psl.noaa.gov/data/correlation/sahelrain.data |
| Southern Oscillation Index | SOI | The SOI is calculated from the air pressure difference between Tahiti (central Pacific) and Darwin, Australia (western Pacific). Positive SOI values indicate La Niña conditions with high pressure over Tahiti and low pressure over Darwin, often linked to increased rainfall in the Sahel. Negative SOI values correspond to El Niño conditions, typically associated with drier Sahel conditions and shifts in global climate patterns. | https://psl.noaa.gov/data/correlation/soi.data |
| Solar Flux | - | Solar flux measures the amount of solar energy reaching Earth, reflecting solar activity cycles. Increased solar flux can lead to warming and changes in atmospheric circulation, which may influence rainfall patterns in Africa. Lower solar flux periods tend to coincide with cooler and more stable climate conditions, potentially affecting the Sahel rainfall variability indirectly. | https://psl.noaa.gov/data/correlation/solar.data |
| Tropical Northern Atlantic Index (TNA) | TNA | The TNA measures sea surface temperature anomalies in the tropical North Atlantic (5°N–25°N, 15°W–60°W). Positive phases with warmer SSTs are associated with increased rainfall along the West African coast and enhanced tropical cyclone activity. Negative phases correspond to cooler SSTs, reduced tropical rainfall, and weaker cyclone activity, often linked to drought conditions in the Sahel. | https://psl.noaa.gov/data/correlation/tna.data |
| Trans Nino Index | TNI | The TNI analyzes spatial shifts in El Niño and La Niña events by measuring SST differences between the eastern tropical Pacific (Niño-1+2) and central tropical Pacific (Niño-4). Positive TNI phases indicate eastward-shifted El Niño effects, increasing rainfall in the eastern Pacific and often suppressing rainfall in the Sahel. Negative phases reflect eastward shifted La Niña effects, which can enhance Sahel precipitation by influencing tropical atmospheric circulation. | https://psl.noaa.gov/data/correlation/tni.data |
| Tropical Southern Atlantic Index | TSA | The TSA measures SST anomalies in the tropical South Atlantic (0°–20°S, 10°E–30°W). Warmer SSTs in the positive phase lead to increased rainfall along eastern South America and shifts in the Atlantic Hadley circulation, which can influence West African monsoon intensity. Cooler SSTs during the negative phase are linked to drought and reduced convection, potentially weakening Sahel rainfall. | https://psl.noaa.gov/data/correlation/tsa.data |
| Tropical Western Hemisphere warm pool | WHWP | The WHWP covers the Caribbean, Gulf of Mexico, and eastern tropical Pacific where SSTs exceed 28°C. Positive phases are characterized by increased temperatures, leading to stronger tropical cyclone activity and enhanced rainfall in surrounding regions. Negative phases correspond to cooler SSTs and reduced storm intensity. WHWP variability affects Atlantic atmospheric circulation and can modulate rainfall in the Sahel and West Africa. | https://psl.noaa.gov/data/correlation/whwp.data |

| West Pacific Index | WPI | The WPI measures atmospheric pressure differences in the tropical and subtropical western Pacific. In its positive phase, a strong high-pressure system weakens Asian monsoons and tropical cyclone activity. The negative phase, dominated by low pressure, enhances Asian monsoon strength and storm activity. Changes in the WPI influence tropical climate dynamics and can indirectly affect the Sahel by modulating global atmospheric circulation patterns. | https://psl.noaa.gov/data/correlation/wp.data |
|---|---|---|---|

**Minor comments:**

1. Abstract line 15, Why should a positive correlation necessarily imply a stronger influence on regional hydrology? Drought is part of hydrology as well; as long as a statistically significant relationship exists—positive or negative—it can affect the system.

   Thanks for the comment. The Authors agree that both positive and negative statistically significant correlations can influence regional hydrology, including drought characteristics. In the revised abstract, we have clarified that the positive correlation with AMO does not necessarily imply a "stronger" influence in general but rather highlights a different type of influence on drought variability compared to indices with negative correlations:

   *Conversely, the Atlantic Multidecadal Oscillation (AMO, 0.40) showed a positive correlation, suggesting its distinct role in modulating hydrological conditions in the Sahel.*

2. Lines 19-20, "further highlights … the NTA" is confusing because the NTA is not introduced earlier.

   Thanks for the comment. To improve clarity, we now introduce the NTA earlier in the abstract along with the other climatic indices considered in the study and mentioned in the Abstract:

   *This study explores the correlation between the Standardized Precipitation Evapotranspiration Index (SPEI) and multiple climatic indices—including the Global Mean Temperature (GMT), Indo-Pacific Warm Pool (IPWP), Atlantic Multidecadal Oscillation (AMO), and North Tropical Atlantic Index (NTA)—using trend analysis, cross-correlation, and an innovative SHAP-driven clustering approach.*

3. Line 21, the abstract does not explain why or how the AI component is explainable.

   Thanks for the comment. The text has been revised to clarify how explainability is achieved through the SHAP framework:

   *The SHAP-driven clustering approach integrates a Random Forest (RF) model with SHAP values to identify distinct drought patterns across the Sahel. By quantifying the contribution of each climatic index to the clustering results, this method makes the model's decision-making process transparent and highlights the prominent influence of AMO and NTA on regional drought variability.*

4. Line 38-39, Gleeson et al. (2012) do not discuss temperature effects. Please check the citation or replace with a more appropriate reference.

Thanks for the comment. Accordingly, the citation has been replaced with more appropriate references that specifically discuss temperature-driven impacts on drought severity and groundwater depletion:

Hao, Z., Hao, F., Singh, V.P. and Zhang, X. (2018). Changes in the severity of compound drought and hot extremes over global land areas. Environmental Research Letters, 13, 124022. doi: 10.1088/1748-9326/aaee96

Nigatu, Z.M., Fan, D., You, W. et al. (2022). Crop production response to soil moisture and groundwater depletion in the Nile Basin based on multi-source data. Science of The Total Environment, 825, 154007. doi: 10.1016/j.scitotenv.2022.154007

Additionally, Gleeson et al. (2012) has been retained in the Introduction but relocated to a more appropriate context: *This over-extraction, coupled with diminished recharge opportunities due to shifting precipitation patterns, raises serious concerns about long-term water security and the sustainability of groundwater-dependent ecosystems (Döll and Fiedler 2008; Gleeson et al. 2012).*

5. Fig. 1 Consider overlaying Köppen climate-type boundaries (or another climate-zone map). This would help readers see whether algorithm-identified clusters align with known climatic regions.

Thanks for the comment. The Köppen-Geiger climate classification has been added to Figure 1.

[Figure]

**Figure 1: Location of the selected SPEI grid in the Sahel region with the Köppen-Geiger Climate Classification.**

6. Lines 134-135, I don't understand how "The 12-month period is long enough to capture the cumulative effect of these global drivers". AMO operates on multi-decadal scales, much longer than 12 months.

Thanks for the comment. The point regarding the temporal scale of indices such as the AMO is well taken. The original statement has been revised to clarify that the 12-month SPEI is not intended to resolve the internal variability of long-term climate modes like the AMO. Rather, the 12-month timescale was selected to capture the integrated hydrological response to prevailing climate conditions over an annual cycle, including the influence of large-scale drivers during their active phases.

While the AMO operates on multidecadal timescales, its current phase (e.g., warm or cool) can influence precipitation patterns in the Sahel on interannual to seasonal scales. The revised text reflects this clarification, emphasizing that the 12-month SPEI captures the expression of such long-term drivers as they affect regional hydroclimatic conditions within a given year.

The revised paragraph has been updated accordingly in the manuscript:

*The 12-month SPEI timescale was selected to reflect both seasonal and interannual climate variability, enabling the detection of annual hydrological responses to the prevailing phases of large-scale climate drivers. While some indices, such as the AMO, operate on multidecadal timescales, their current phase can still exert influence on precipitation patterns within a given year. Thus, the 12-month period is not intended to resolve long-term climate variability itself, but rather to integrate its effects as expressed in a single year's climate system.*

7. Lines 140-143, The logic is hard to follow. Clarify why having 31 indices conflicts with a 1951–2018 record, and why a "large number of indices" would undermine a robust analysis. Re-phrase to make the trade-offs explicit.

Thanks for the comment. The original statement aimed to address the potential challenge of integrating a large number of climate indices—each with varying periods of availability—within a coherent analytical framework. While the inclusion of 31 indices increases the risk of reduced temporal overlap, multicollinearity, and noise in the analysis, the 1951–2018 period was selected because it represents the longest continuous span with consistent data coverage for the majority of indices considered.

This timeframe ensures sufficient temporal overlap across most datasets, allowing for a methodologically robust analysis of long-term climatic variability and its relationship with drought. The revised text now explicitly outlines these trade-offs and justifies the selection of both the number of indices and the chosen time period

The revised paragraph has been updated accordingly in the manuscript:

*Although the analysis incorporated 31 climate indices, the historical period from 1951 to 2018 was deemed appropriate for this study. This timeframe balances the need for a sufficiently long record to capture long-term*

*climatic variability with the availability of consistent and overlapping data for a large set of indices. While the inclusion of many indices can pose challenges—such as reduced temporal overlap, increased multicollinearity, and potential noise in statistical relationships—the 1951–2018 period provided a common baseline that ensured temporal consistency across most indices. As a result, it was possible to conduct a robust analysis of long-term climate-drought relationships while minimizing data limitations associated with differing index availability.*

8. Fig. 3a, I suggest adding hatches or stipples to distinguish areas with and without statistically significant trends. Same thing for Fig. 5.

Thanks for the suggestion. For Figures 3a and 5, a more distinct color bar has been used to better differentiate the various ranges of Z-values. Also Figure 6 and 7 have been improved to better differentiate colors.

Regarding the use of hatches or stipples, it should be noted that the grid is very dense, with all cells fully populated. Adding such patterns would substantially reduce the readability of the map and may obscure underlying spatial structures. For this reason, color intensity was intentionally used as the primary visual cue for statistical significance.

[Figure]

**Figure 3: Z parameter of the SK test: SPEI-12 map (a)**

[Figure]

**SPEI-12 - CLIMATIC INDICES - Correlation coefficient (r)**

■ -1.00 - -0.50 ■ -0.50 - -0.25 ■ -0.25 - 0 ■ 0 - 0.25 ■ 0.25 - 0.50 ■ 0.50 - 1.00

[Figure]

**Figure 5. Maps of the correlations between SPEI-12 gridded data and the most correlated climatic indices (continue).**

[Figure]

**Figure 6.** Correlation analysis between SPEI-12 for Cell 2042 and AMO. The figure presents the time series of AMO and SPEI-12 for Cell 2042, located at the border between Chad and Sudan. Additionally, it includes a scatter plot illustrating their relationship on both a monthly scale and a five-year mean scale.

[Figure]

**Figure 7. Correlation analysis between SPEI-12 for Cell 2319 and GMT. The figure presents the time series of AMO and SPEI-12 for Cell 2319, located at the Sahel's border in Central Sudan. Additionally, it includes a scatter plot illustrating their relationship on both a monthly scale and a five-year mean scale.**

9. Line 210-219, When discussing the impact of climate variability on drought, indicate the direction of influence. For example, does increased aerosol loading tend to increase or decrease regional precipitation?

Thanks for the comment. The paragraph discussing the results of the Seasonal Kendall test on the climate indices has been revised to explicitly state the direction of influence each trend is likely to exert on regional precipitation and drought conditions. In particular, the revised text now clarifies that increases in GMT and IPWP are generally

associated with reduced rainfall and enhanced drought conditions in the Sahel. Likewise, the decreasing trends in TNI, Solar Flux, and Sahel P are discussed in terms of their potential to contribute to regional drying, with reduced solar flux—possibly linked to increased aerosol loading or cloud cover—being associated with suppressed precipitation. These clarifications aim to provide a clearer understanding of how specific climate signals relate to observed hydroclimatic changes in the region. Revised text:

*The SK test was also performed for the climatic indices (Figure 3b). The predominance of statistically significant increasing trends, particularly for the IPWP (Z = 27.83) and GMT (Z = 28.70), underscores the substantial role of global warming and oceanic heat distribution in shaping regional climate dynamics. These upward trends reflect broader increases in sea surface temperatures and global temperature anomalies, which are generally associated with reduced precipitation and enhanced drought conditions in the Sahel due to shifts in atmospheric circulation and moisture availability.*

*Conversely, statistically significant decreasing trends were observed in three indices: TNI (Z = -7.83), Solar Flux (Z = -3.18), and Sahel P (Z = -4.13), each suggesting mechanisms that contribute to regional drying. The decline in TNI implies a weakening of tropical convection and changes in atmospheric circulation patterns that can reduce moisture transport toward the Sahel. The decrease in Solar Flux may be indicative of increased aerosol concentrations or cloud cover, both of which tend to reduce surface solar radiation, leading to lower evaporation and altered atmospheric dynamics that often result in reduced rainfall. Finally, the negative trend in Sahel P reflects a direct decline in regional precipitation, consistent with the observed intensification and persistence of drought conditions in recent decades.*

10. Line 266, Define the threshold for "weaker correlations" and state the correlation values, not just the IQR.

Thanks for the comment. The text has been updated to report both the IQR and the mean correlation values for each index discussed, thereby providing a clearer quantitative basis for interpretation. The term "weaker correlations" refers to indices such as AO (mean = -0.02, IQR = 0.05) and NAO (mean = -0.03, IQR = 0.06), which exhibit notably lower correlation magnitudes and narrower variability compared to other indices in the analysis. This classification is intended to highlight their comparatively limited and less consistent relationship with drought variability in the Sahel, relative to indices such as AMM, AMO, and GMT.

11. Line 341, what statistic of SHAP values do we use to measure the influence on clustering? I thought I should look at the mean values but here the authors cite the range.

Thanks for the comment. The influence of each climatic index on the clustering outcome is assessed by examining the distribution of SHAP values assigned to that feature across all data points. In the revised manuscript, more details on the SHAP analysis and on the Random Forest model have been provided.

As an example of the text, here you find the revised version of the description for cluster C1:

*In Cluster C1, the AMO, CAR and TNA emerged as the most influential variables, with mean absolute SHAP values of 0.088, 0.72 and 0.059, respectively (see Table S5). Their distributions are notably skewed toward positive SHAP values, with dense concentrations between 0.05 and 0.15. This pattern indicates a strong and consistent association between high index values and increased likelihood of C1 classification. Moderately influential indices such as WHWP, NTA, and AMM present narrower spreads (-0.05 to 0.2) and more symmetric profiles, suggesting subtler but still directional contributions. Conversely, indices like PDO, GMT, IPWP, and TSA show very limited SHAP influence, with values clustered near zero and minimal dispersion, highlighting their negligible role in defining this cluster.*

In addition, in the Supplementary material, Tables S4 and S5 provide the Output accuracy of the Random Forest model and the SHAP Feature Importance Ranking with Confidence Intervals, respectively.

**Table S4. Output accuracy of the Random Forest model. The color bar ranges from red (low values) to green (high values).**

| Clusters | precision | recall | f1-score | support |
|---|---|---|---|---|
| C1 | 1 | 0.97 | 0.985 | 33 |
| C2 | 0.964 | 1 | 0.982 | 54 |
| C3 | 1 | 0.978 | 0.989 | 46 |
| accuracy | | | 0.985 | 133 |
| macro average | 0.988 | 0.983 | 0.985 | 133 |
| weighted average | 0.985 | 0.985 | 0.985 | 133 |
| Overall Random Forest Accuracy | 0.985 | | | |

**Table S5. SHAP Feature Importance Ranking with Confidence Intervals. The color bar ranges from red (low values) to green (high values).**

| Classe | Feature | Mean Absolute SHAP values | Lower 95% Confidence Interval | Upper 95% Confidence Interval |
|---|---|---|---|---|
| C1 | AMO | 0.088 | 0.079 | 0.098 |
| | CAR | 0.072 | 0.064 | 0.081 |
| | TNA | 0.059 | 0.051 | 0.066 |
| | WHWP | 0.051 | 0.045 | 0.057 |
| | NTA | 0.042 | 0.038 | 0.048 |
| | AMM | 0.033 | 0.029 | 0.037 |
| | Sahel P | 0.013 | 0.011 | 0.014 |
| | PDO | 0.009 | 0.008 | 0.011 |
| | GMT | 0.006 | 0.005 | 0.006 |
| | IPWP | 0.002 | 0.002 | 0.002 |

| | | | | |
|---|---|---|---|---|
| | TSA | 0.002 | 0.002 | 0.002 |
| C2 | AMO | 0.096 | 0.091 | 0.101 |
| | NTA | 0.084 | 0.079 | 0.088 |
| | CAR | 0.077 | 0.072 | 0.081 |
| | TNA | 0.067 | 0.063 | 0.072 |
| | WHWP | 0.061 | 0.056 | 0.066 |
| | IPWP | 0.029 | 0.026 | 0.033 |
| | AMM | 0.027 | 0.025 | 0.030 |
| | GMT | 0.023 | 0.020 | 0.027 |
| | Sahel P | 0.014 | 0.013 | 0.016 |
| | PDO | 0.009 | 0.008 | 0.011 |
| | TSA | 0.007 | 0.006 | 0.008 |
| C3 | NTA | 0.102 | 0.094 | 0.109 |
| | AMO | 0.080 | 0.073 | 0.087 |
| | WHWP | 0.072 | 0.067 | 0.078 |
| | CAR | 0.061 | 0.057 | 0.066 |
| | TNA | 0.059 | 0.055 | 0.064 |
| | IPWP | 0.030 | 0.027 | 0.033 |
| | GMT | 0.027 | 0.024 | 0.030 |
| | AMM | 0.012 | 0.011 | 0.014 |
| | TSA | 0.006 | 0.006 | 0.007 |
| | Sahel P | 0.004 | 0.003 | 0.004 |
| | PDO | 0.002 | 0.002 | 0.002 |

12. Line 344, "High" and "low" should be replaced with actual correlation values (or value ranges). Note that Fig. 9 labels "feature value," not "correlation."

Thanks for the comment. The text has been revised to specify that the color gradient in Figure 9 represents the actual correlation values between each climatic index and SPEI-12, with the range clearly indicated from low (blue) to high (red). Revised text:

*In the SHAP beeswarm plots (Figure 9), the x-axis represents SHAP values—the impact of each feature on the clustering outcome—while the color gradient (Feature value) encodes the actual correlation value between the climatic index and SPEI-12 for each data point, ranging from low (blue) to high (red). This dual encoding enables a nuanced interpretation of the model's behavior: the position along the x-axis reflects the strength and direction of influence, while the color reveals whether strong or weak correlations drive the effect.*

13. Lines 475-481, Spatial heterogeneity has already been discussed in lines 415-419. Avoid repetition.

Thank you for this observation. The text in lines 473–481 has been revised to avoid redundancy with the earlier discussion on spatial heterogeneity in lines 415–419. The revised paragraph now focuses more specifically on the added value of SHAP analysis in identifying the relative importance of individual climatic indices, particularly the

consistent influence of the NTA, without repeating the earlier interpretation of AMO-related spatial variability. This adjustment preserves the integrity of the findings while improving clarity and conciseness:

*In this study, the application of SHAP analysis provided insight into the relative importance of individual climate indices in shaping the clustering structure. Indices such as AMO and NTA emerged as influential in distinguishing cluster-specific drought patterns. Notably, the NTA consistently showed a negative correlation with SPEI-12, suggesting that cooler SSTs in this region are associated with wetter conditions. These results reinforce the role of oceanic variability in modulating drought conditions and demonstrate the added value of interpretable machine learning methods in identifying key drivers of regional differentiation without assuming uniform climatic influence.*

14. Lines 512-519, Link the limitation of ignoring human activities to specific findings—e.g., could regions with low climate–SPEI correlation coincide with areas of extensive land-use change or other human activities?

Thank you for this constructive suggestion. The paragraph discussing methodological limitations has been revised to explicitly acknowledge that regions exhibiting weak correlations between climate indices and SPEI-12 may correspond to areas affected by significant anthropogenic influences, such as land-use change, irrigation, or groundwater extraction. This clarification reinforces the relevance of extending the analysis to highly anthropized environments, where non-climatic drivers may decouple local drought dynamics from broader climate variability. The revised text also emphasizes the potential value of applying the methodology to such contexts to better understand the interaction between human-induced modifications and large-scale climatic controls:

*Additionally, while this study successfully integrates trend analysis and explainable clustering, further investigation is needed to assess its performance in highly anthropized environments, where urbanization, land-use change, and water extraction exert non-climatic controls on drought evolution. Notably, some areas showing weak correlations between climate indices and SPEI-12 may coincide with regions undergoing extensive human-induced modifications, such as agricultural expansion, irrigation, or groundwater exploitation. These anthropogenic factors can decouple local drought dynamics from large-scale climate drivers, potentially obscuring the climate signal detected by statistical models. Expanding the application of this methodology to regions experiencing rapid demographic growth and infrastructural development, such as peri-urban zones increasingly reliant on groundwater, would offer critical insights into the interplay between human activities and climatic variability. Similarly, applying the approach to colder climates would enable an evaluation of its robustness in regions where snowpack dynamics, freeze-thaw processes, and permafrost degradation introduce additional layers of hydrological complexity.*

---

## Author Comment (AC2)

**Reviewer 2**

The manuscript titled *"Decoding the Architecture of Drought: SHAP-Enhanced Insights into the Climate Forces Reshaping the Sahel"* presents a robust, interdisciplinary analysis of drought patterns in the Sahel region. The authors employ a multi-method approach that combines the Standardized Precipitation Evapotranspiration Index (SPEI), Seasonal Kendall (SK) trend analysis, cross-correlation with 31 climatic indices, and a SHAP-enhanced clustering methodology using Random Forest (RF) to explore the spatial-temporal variability of drought and its climatic drivers.

Key findings include:

- A significant downward trend in SPEI-12 across 57.5% of the Sahel, particularly in the west and southeast, indicating intensified drought conditions.

- Strong negative correlations between drought severity and Global Mean Temperature (GMT) and Indo-Pacific Warm Pool (IPWP); Atlantic Multidecadal Oscillation (AMO) showed spatially heterogeneous impacts.

- The clustering analysis delineates three distinct regions with unique drought dynamics and climate-drought interactions.

- The SHAP framework reveals the differential contribution of climatic indices to drought clustering, offering high interpretability and novel insight into region-specific vulnerabilities.

We sincerely thank the Reviewer for the thoughtful and thorough evaluation of our manuscript. The manuscript has been revised in accordance with all comments received. Additional modifications have also been made during the revision process, which have contributed to improving the overall quality and clarity of the text. It is hoped that the revised version will be found suitable for publication. A point-by-point response to the Reviewer's comments is provided below.

**Title and Abstract**

- Include quantitative results (e.g., number of clusters, correlation values) in the abstract to enhance clarity and impact.

- Slightly reduce jargon in the abstract for broader accessibility (e.g., explain "SHAP" in simpler terms before the acronym).

Thanks for the comment. The Abstract has been improved including quantitative results and slightly reducing jargon in the abstract for broader accessibility.

**Introduction**

- Include a short paragraph summarizing existing clustering approaches and why SHAP-RF is a significant improvement.

- Reduce the length of some paragraphs to improve readability and flow.

The Introduction has been revised by shortening several paragraphs to enhance readability and flow, and by adding a paragraph summarizing existing clustering approaches and explaining why the SHAP framework represents a significant improvement:

*A critical yet frequently overlooked aspect of drought characterization involves identifying spatially homogeneous regions that exhibit consistent drought-climate relationships. Traditionally, clustering techniques such as K-means and Hierarchical clustering have been used to delineate these regions based on hydroclimatic features. K-means, while computationally efficient, assumes spherical clusters and equal variance, often oversimplifying complex spatial patterns. Hierarchical clustering, although more flexible in capturing nested relationships, can be sensitive to noise and lacks scalability for large datasets. Moreover, both methods operate as unsupervised learning algorithms, providing little insight into the underlying climatic drivers that influence cluster formation. As a result, these techniques often fall short in interpretability and in explaining the climatic processes shaping spatial drought variability.*

*To overcome these limitations, this study introduces an innovative SHAP-driven clustering framework, which integrates RF classification with SHAP analysis. In this approach, RF is used to classify observations into drought-prone clusters identified during the unsupervised phase, while SHAP quantifies the contribution of each climatic variable to the predicted cluster membership. This combination offers a transparent and interpretable alternative to traditional clustering by uncovering not only the spatial patterns of drought but also the relative importance of different climate drivers in shaping those patterns. The framework shifts from a purely data-partitioning paradigm to one that integrates explainable AI, significantly enhancing the understanding of how climatic variability governs regional drought dynamics.*

**Materials and Methods**

- Consider summarizing the 31 climate indices in a supplementary table only, instead of the main text, or condensing Table 1.

    Thanks for the comment. Table 1 has been updated to include a more concise descriptions of each index, highlighting their potential impacts on Africa and the Sahel where relevant, and the data source.

| Climate index | Abbr. | Definition | Data source |
|---|---|---|---|
| Atlantic Meridional Mode | AMM | The AMM describes north-south SST differences in the tropical Atlantic. Its positive phase shifts rainfall northward, increasing Sahel precipitation and reducing drought risk. The negative phase causes southward rainfall shifts, | https://psl.noaa.gov/data/timeseries/month/DS/AMM/ |

| | | | |
|---|---|---|---|
| | | leading to Sahel drought. AMM also affects Atlantic hurricane activity, influencing regional climate variability. | |
| Atlantic Multidecadal Oscillation | AMO | The Atlantic Multidecadal Oscillation refers to natural variations in North Atlantic Ocean sea surface temperatures that occur over periods of 20 to 40 years. In its positive phase, North Atlantic temperatures are above average, leading to hotter summers along the eastern U.S., increased hurricane activity in the tropical Atlantic, and enhanced rainfall in Africa. In its negative phase, cooler Atlantic temperatures are associated with weaker hurricane activity, drought in Africa's Sahel region, and cooler, wetter summers in Europe. The AMO plays a significant role in shaping global climate systems and regional weather patterns, particularly in the North Atlantic region. | https://www.psl.noaa.gov/data/timeseries/AMO/ |
| Arctic Oscillation | AO | The AO influences atmospheric circulation patterns that can extend to the Sahel region by affecting the strength and position of the African Easterly Jet and mid-latitude weather systems. Its negative phase can weaken the jet stream, altering rainfall patterns in West Africa and contributing to Sahel drought or variability in seasonal precipitation. | https://psl.noaa.gov/data/timeseries/month/DS/AO/ |
| Berkeley Earth Surface Temperature | BEST | The BEST dataset provides global and regional surface temperature trends, including detailed temperature anomalies across Africa. These trends are crucial for understanding how warming influences Sahel hydroclimate, as rising temperatures can exacerbate drought conditions and impact rainfall variability in the region. | https://psl.noaa.gov/data/correlation/censo.data |
| Caribbean Index | CAR | The CAR index captures climate variability in the Caribbean, including SST and atmospheric patterns. It influences Atlantic tropical cyclone activity, which can affect West African monsoon dynamics and Sahel rainfall through atmospheric teleconnections. | https://psl.noaa.gov/data/correlation/CAR_ersst.data |
| Eastern Pacific Oscillation Index | EPO | The EPO describes atmospheric pressure anomalies in the eastern North Pacific. Its phases influence the jet stream and temperature patterns in North America, which can indirectly affect West African climate by modulating large-scale atmospheric circulation and teleconnections linked to Sahel rainfall variability. | https://psl.noaa.gov/data/correlation/epo.data |
| Greenland Blocking Index | GBI | The GBI measures persistent high-pressure systems over Greenland. Its positive phase alters North Atlantic circulation, which can influence the West African monsoon and Sahel rainfall by affecting atmospheric patterns that modulate moisture transport into the region. | https://psl.noaa.gov/data/correlation/gbi.ncep.day |
| Global Mean Temperature | GMT | GMT tracks overall atmospheric and ocean warming or cooling trends. Rising global temperatures influence the Sahel by intensifying droughts, altering rainfall patterns, and impacting regional water resources through shifts in the hydrological cycle. | https://psl.noaa.gov/data/correlation/gmsst.data |
| Indo-Pacific Warm Pool | IPWP | The IPWP, with some of the warmest tropical ocean temperatures, drives global atmospheric circulation, including monsoons. Its warming phase enhances convection and rainfall, indirectly influencing Sahel rainfall through shifts in the Walker circulation and global moisture transport. | https://psl.noaa.gov/data/correlation/pacwarm.data |
| The North Atlantic Oscillation | NAO | The NAO index measures sea-level pressure differences between the Azores High and the Subpolar Low. Its phases modulate the North Atlantic jet stream and storm tracks, affecting heat and moisture transport. These changes influence West African monsoon strength and Sahel precipitation by altering atmospheric circulation patterns over the Atlantic. | https://psl.noaa.gov/data/correlation/nao.data |
| The North Atlantic Oscillation (Jones) | NAO (Jones) | Defined by Jones (1997), this NAO index measures the sea-level pressure difference between the Azores High and Icelandic Low. Its phases influence Atlantic atmospheric circulation patterns that affect West African monsoon dynamics and Sahel rainfall variability. | https://psl.noaa.gov/data/correlation/jonesnao.data |
| Niño-1.2 | - | This index covers sea surface temperatures in the eastern equatorial Pacific (80°W–90°W, 10°S–0°), where El Niño and La Niña events typically originate. In its positive phase (El Niño), warmer waters lead to increased rainfall and floods along South America's northwest coast; in its negative phase (La Niña), cooler waters cause drought and promote cold-water upwelling. While its direct effects are regional, Niño-1+2 influences large-scale atmospheric circulation patterns, which can alter the West African monsoon strength and consequently affect Sahel precipitation variability. | https://www.cpc.ncep.noaa.gov/data/indices/ersst5.nino.mth.91-20.ascii |
| Niño 3 | - | The Niño-3 index tracks sea surface temperatures in the eastern equatorial Pacific (150°W–90°W, 5°S–5°N) to monitor El Niño and La Niña events. | https://www.cpc.ncep.noaa.gov/data/ |

| | | | |
|---|---|---|---|
| | | During El Niño (positive phase), warmer waters cause increased rainfall in western South America, drought in Asia-Pacific, and reduced rainfall in the Sahel. La Niña (negative phase) brings cooler waters, increased storms in Asia-Pacific, and enhanced rainfall in the Sahel. This index influences global atmospheric circulation and tropical rainfall patterns. | indices/ersst5.nino.mth.91-20.ascii |
| Niño-3.4 | - | The Niño-3.4 index measures sea surface temperatures in the central equatorial Pacific (120°W–170°W, 5°S–5°N) and is a key indicator of El Niño and La Niña events. During El Niño (positive phase), warmer waters lead to increased rainfall along South America's coast, drought and heatwaves in the Asia-Pacific, and reduced rainfall in the Sahel. La Niña (negative phase) brings cooler waters, heavy rains and floods in Asia-Pacific, drought in South America, and enhanced rainfall in the Sahel. Niño-3.4 strongly influences global atmospheric circulation and tropical weather patterns. | https://www.cpc.ncep.noaa.gov/data/indices/ersst5.nino.mth.91-20.ascii |
| Niño-4 | - | The Niño-4 index measures sea surface temperature variations in the central Pacific (160°E–150°W, 5°S–5°N) during El Niño and La Niña events. In its positive phase (El Niño), warmer waters enhance convection in the tropical western Pacific, influencing atmospheric circulation and monsoon patterns, often linked to reduced rainfall in the Sahel. In the negative phase (La Niña), cooler waters lead to drought in the western Pacific and increased rainfall in the central Pacific, sometimes boosting Sahel precipitation. Niño-4 is key for understanding tropical circulation and regional climate variability. | https://www.cpc.ncep.noaa.gov/data/indices/ersst5.nino.mth.91-20.ascii |
| Northern Oscillation Index | NOI | The NOI measures sea level pressure differences between Tahiti (eastern tropical Pacific) and Darwin (western subtropical Pacific). In its positive phase, high pressure dominates the eastern Pacific and low pressure the western Pacific, causing drought in the east and wetter conditions in the west. The negative phase reverses this pattern. The NOI is essential for understanding tropical climate phenomena such as El Niño-Southern Oscillation (ENSO) impacts. | https://psl.noaa.gov/data/correlation/noi.data |
| North Pasific Index | NP | The NP reflects average sea level pressure over the North Pacific, indicating the strength of the Aleutian Low. In its positive phase, a stronger Aleutian Low brings more rainfall to western North America and cooler eastern Pacific waters. In the negative phase, the system weakens, leading to drier conditions and warmer sea surface temperatures. The NP Index is key to understanding Pacific atmospheric circulation and its effects on North American and, to a lesser extent, African climate patterns | https://psl.noaa.gov/data/correlation/np.data |
| North Tropical Atlantic SST Index | NTA | The NTA index represents SSTs in the North Tropical Atlantic. In its positive phase, warmer SSTs enhance convection and tropical cyclone activity, while influencing rainfall patterns across the Atlantic basin. This phase is often linked to increased precipitation in the Sahel. In the negative phase, cooler SSTs reduce cyclone activity and can lead to drought conditions in West Africa. The NTA index is crucial for understanding Atlantic climate variability and its impacts on regional weather systems. | https://psl.noaa.gov/data/correlation/NTA_ersst.data |
| Oceanic Niño Index | ONI | The ONI measures the three-month running average of sea surface temperature anomalies in the Niño-3.4 region of the central tropical Pacific. It is the primary indicator of El Niño and La Niña events within the ENSO cycle. Positive ONI values (El Niño) are linked to drought in regions like Australia and the Sahel, while negative values (La Niña) can enhance rainfall in these areas. The ONI is essential for monitoring ENSO's global impacts on temperature, rainfall, and atmospheric circulation | https://psl.noaa.gov/data/correlation/oni.data |
| Pasific Decadal Oscillation | PDO | The PDO describes long-term shifts in sea surface temperatures and atmospheric pressure across the North Pacific. In its positive phase, the eastern Pacific cools while the western Pacific warms, bringing wetter conditions to western North America and warmer weather in Alaska. In the negative phase, the pattern reverses, causing drought in western North America and reduced marine productivity. The PDO influences multi-decadal climate variability, affecting agriculture, fisheries, and water resources globally, including rainfall patterns in Africa | https://psl.noaa.gov/data/correlation/pdo.data |
| Pacific Meridional Mode | PMM | The PMM is a climate pattern driven by interactions between sea surface temperatures and surface winds in the tropical Pacific. In its positive phase, warmer waters enhance convection and rainfall across the Pacific, often preconditioning El Niño events. In its negative phase, cooler waters suppress | https://psl.noaa.gov/data/timeseries/month/data/pmm.data |

| | | convection, leading to drier tropical conditions. The PMM influences the onset of ENSO events and plays a key role in shaping tropical and global climate variability, including rainfall over Africa. | |
|---|---|---|---|
| Pacific–North American Pattern | PNA | The PNA pattern is a major mode of atmospheric variability in the Northern Hemisphere, reflecting recurring pressure anomalies over the North Pacific and North America. The PNA index is based on standardized 500 hPa geopotential height anomalies at four specific locations. Its phases correlate with temperature and precipitation anomalies across North America. The PNA influences regional weather by modulating the strength and position of the East Asian jet stream, affecting storm tracks and climate patterns. Through atmospheric teleconnections, the PNA can also impact tropical circulations, with potential links to rainfall variability in the Sahel and parts of northern Africa. | https://psl.noaa.gov/data/correlation/pna.data |
| Quasi-Biennial Oscillation | QBO | The QBO is a regular oscillation of easterly and westerly winds in the tropical stratosphere, with a cycle of about 28–30 months. In its westerly phase, tropical cyclone activity increases, especially in the Pacific and Atlantic. In the easterly phase, cyclone formation weakens, and stratospheric ozone distribution shifts. The QBO modulates stratosphere–troposphere interactions and can influence tropical convection, potentially affecting rainfall variability in regions such as the Sahel and equatorial Africa. | https://psl.noaa.gov/data/correlation/qbo.data |
| Sahel Precipitation | Sahel P | Sahel Precipitation refers to the annual rainfall in Africa's Sahel region (south of the Sahara Desert) and is influenced by tropical Atlantic Sea surface temperatures and atmospheric circulation. Positive phases (increased rainfall) improve agriculture and water resources, while negative phases (drought) lead to famine and heightened socio-economic impacts. | https://psl.noaa.gov/data/correlation/sahelrain.data |
| Southern Oscillation Index | SOI | The SOI is calculated from the air pressure difference between Tahiti (central Pacific) and Darwin, Australia (western Pacific). Positive SOI values indicate La Niña conditions with high pressure over Tahiti and low pressure over Darwin, often linked to increased rainfall in the Sahel. Negative SOI values correspond to El Niño conditions, typically associated with drier Sahel conditions and shifts in global climate patterns. | https://psl.noaa.gov/data/correlation/soi.data |
| Solar Flux | - | Solar flux measures the amount of solar energy reaching Earth, reflecting solar activity cycles. Increased solar flux can lead to warming and changes in atmospheric circulation, which may influence rainfall patterns in Africa. Lower solar flux periods tend to coincide with cooler and more stable climate conditions, potentially affecting the Sahel rainfall variability indirectly. | https://psl.noaa.gov/data/correlation/solar.data |
| Tropical Northern Atlantic Index (TNA) | TNA | The TNA measures sea surface temperature anomalies in the tropical North Atlantic (5°N–25°N, 15°W–60°W). Positive phases with warmer SSTs are associated with increased rainfall along the West African coast and enhanced tropical cyclone activity. Negative phases correspond to cooler SSTs, reduced tropical rainfall, and weaker cyclone activity, often linked to drought conditions in the Sahel. | https://psl.noaa.gov/data/correlation/tna.data |
| Trans Nino Index | TNI | The TNI analyzes spatial shifts in El Niño and La Niña events by measuring SST differences between the eastern tropical Pacific (Niño-1+2) and central tropical Pacific (Niño-4). Positive TNI phases indicate eastward-shifted El Niño effects, increasing rainfall in the eastern Pacific and often suppressing rainfall in the Sahel. Negative phases reflect eastward shifted La Niña effects, which can enhance Sahel precipitation by influencing tropical atmospheric circulation. | https://psl.noaa.gov/data/correlation/tni.data |
| Tropical Southern Atlantic Index | TSA | The TSA measures SST anomalies in the tropical South Atlantic (0°–20°S, 10°E–30°W). Warmer SSTs in the positive phase lead to increased rainfall along eastern South America and shifts in the Atlantic Hadley circulation, which can influence West African monsoon intensity. Cooler SSTs during the negative phase are linked to drought and reduced convection, potentially weakening Sahel rainfall. | https://psl.noaa.gov/data/correlation/tsa.data |
| Tropical Western Hemisphere warm pool | WHWP | The WHWP covers the Caribbean, Gulf of Mexico, and eastern tropical Pacific where SSTs exceed 28°C. Positive phases are characterized by increased temperatures, leading to stronger tropical cyclone activity and enhanced rainfall in surrounding regions. Negative phases correspond to cooler SSTs and reduced storm intensity. WHWP variability affects Atlantic atmospheric circulation and can modulate rainfall in the Sahel and West Africa. | https://psl.noaa.gov/data/correlation/whwp.data |

| West Pacific Index | WPI | The WPI measures atmospheric pressure differences in the tropical and subtropical western Pacific. In its positive phase, a strong high-pressure system weakens Asian monsoons and tropical cyclone activity. The negative phase, dominated by low pressure, enhances Asian monsoon strength and storm activity. Changes in the WPI influence tropical climate dynamics and can indirectly affect the Sahel by modulating global atmospheric circulation patterns. | https://psl.noaa.gov/data/correlation/wp.data |
|---|---|---|---|

- Include more explanation or citation on how SHAP values are computed and interpreted in the clustering context for readers unfamiliar with explainable AI techniques.

Thanks for the comment. In the revised version of the manuscript, the Methodological Section has been improved providing a more detailed explanation of the SHAP analysis:

*However, this criterion does not allow for a clear assessment of the impact of each climatic index on the clustering process. To enhance the interpretability of clustering analyses in hydrological studies, particularly concerning drought patterns, this study integrates SHAP (values with RF models. This approach addresses the limitations of traditional clustering methods, which often lack explanatory power regarding the influence of individual climatic variables on cluster formation.*

*SHAP, grounded in cooperative game theory, assigns each feature an important value for a particular prediction, offering a unified measure of feature influence across the model. In this study, SHAP values are employed to interpret the output of an RF classifier trained to predict cluster assignments based on climatic indices (Lundberg and Lee 2017). The process involves:*

- *Model Training: An RF classifier is trained using climatic indices as input features and the cluster labels (obtained from initial clustering analyses) as the target variable.*

- *SHAP Value Computation: Post-training, SHAP values are computed for each feature, quantifying the contribution of each climatic index to the model's prediction for each data point. This computation considers all possible combinations of features, ensuring a fair distribution of importance among them .*

- *Interpretation: The resulting SHAP values provide insights into how each climatic index influences the assignment of data points to specific clusters. Positive SHAP values indicate a feature's positive contribution to predicting a particular cluster, while negative values suggest a negative contribution.*

*By employing this SHAP-driven approach, the study transforms clustering from a purely statistical exercise into an interpretable framework that reveals the underlying climatic drivers of drought patterns. This enhanced interpretability facilitates more informed decision-making and targeted adaptation strategies, especially in regions like the Sahel, where drought dynamics are influenced by complex interactions among multiple climatic factors.*

*This methodology aligns with recent advancements in explainable AI, where SHAP values have been utilized to enhance the interpretability of clustering analyses in various domains (Cohen et al. 2023). By integrating SHAP with RF models, the study not only identifies homogeneous drought regions but also elucidates the specific climatic variables driving these patterns, thereby contributing to more effective drought mitigation and resource management strategies.*

**Results**

- Provide statistical significance or validation metrics for SHAP impacts (e.g., confidence intervals or feature importance rankings).

  Thanks for the comment. In supplementary material Table S5 has been reported, providing the SHAP Feature Importance Ranking with the Confidence Intervals. The text in the paper has been also updated accordingly. Below an extract of the updated text:

  *In Cluster C1, the AMO, CAR and TNA emerged as the most influential variables, with mean absolute SHAP values of 0.088, 0.72 and 0.059, respectively (see Table S5). Their distributions are notably skewed toward positive SHAP values, with dense concentrations between 0.05 and 0.15. This pattern indicates a strong and consistent association between high index values and increased likelihood of C1 classification. Moderately influential indices such as WHWP, NTA, and AMM present narrower spreads (-0.05 to 0.2) and more symmetric profiles, suggesting subtler but still directional contributions. Conversely, indices like PDO, GMT, IPWP, and TSA show very limited SHAP influence, with values clustered near zero and minimal dispersion, highlighting their negligible role in defining this cluster.*

  *Cluster C2, in contrast, was characterized by AMO and the NTA indices as the most important features (mean absolute SHAP equal to 0.096 and 0.084, respectively), followed by CAR and TNA. These variables show significant spread on both sides of zero, implying a bidirectional influence where both high and low values can affect classification, depending on the context. Secondary contributors such as IPWP, AMM, and GMT exhibit tighter distributions centered around zero but with occasional asymmetries, pointing to context-dependent roles. Sahel P, PDO, and TSA remain minimally influential, with narrow SHAP ranges and modes at or near zero. Compared to Cluster C1, the SHAP profiles in C2 suggest greater interaction complexity among variables rather than dominance by a few.*

  *For Cluster C3, NTA dominated the feature importance ranking (mean absolute SHAP: 0.102), followed by AMO (0.080) and WHWP (0.072). These distributions are distinctly positively skewed, and the color gradient confirms that high feature values strongly align with positive SHAP contributions. Variables such as CAR, TNA, and IPWP*

*follow a similar, though slightly less pronounced, pattern. Mid-tier contributors like GMT, AMM, and TSA are more symmetrically distributed, with modal SHAP values just above zero. Finally, Sahel P and PDO again register as the least impactful, mirroring the behavior observed in the other clusters.*

**Table S5. SHAP Feature Importance Ranking with Confidence Intervals. The color bar ranges from red (low values) to green (high values).**

| Classe | Feature | Mean Absolute SHAP values | Lower 95% Confidence Interval | Upper 95% Confidence Interval |
|--------|---------|---------------------------|-------------------------------|-------------------------------|
| C1 | AMO | 0.088 | 0.079 | 0.098 |
| | CAR | 0.072 | 0.064 | 0.081 |
| | TNA | 0.059 | 0.051 | 0.066 |
| | WHWP | 0.051 | 0.045 | 0.057 |
| | NTA | 0.042 | 0.038 | 0.048 |
| | AMM | 0.033 | 0.029 | 0.037 |
| | Sahel P | 0.013 | 0.011 | 0.014 |
| | PDO | 0.009 | 0.008 | 0.011 |
| | GMT | 0.006 | 0.005 | 0.006 |
| | IPWP | 0.002 | 0.002 | 0.002 |
| | TSA | 0.002 | 0.002 | 0.002 |
| C2 | AMO | 0.096 | 0.091 | 0.101 |
| | NTA | 0.084 | 0.079 | 0.088 |
| | CAR | 0.077 | 0.072 | 0.081 |
| | TNA | 0.067 | 0.063 | 0.072 |
| | WHWP | 0.061 | 0.056 | 0.066 |
| | IPWP | 0.029 | 0.026 | 0.033 |
| | AMM | 0.027 | 0.025 | 0.030 |
| | GMT | 0.023 | 0.020 | 0.027 |
| | Sahel P | 0.014 | 0.013 | 0.016 |
| | PDO | 0.009 | 0.008 | 0.011 |
| | TSA | 0.007 | 0.006 | 0.008 |
| C3 | NTA | 0.102 | 0.094 | 0.109 |
| | AMO | 0.080 | 0.073 | 0.087 |
| | WHWP | 0.072 | 0.067 | 0.078 |
| | CAR | 0.061 | 0.057 | 0.066 |
| | TNA | 0.059 | 0.055 | 0.064 |
| | IPWP | 0.030 | 0.027 | 0.033 |
| | GMT | 0.027 | 0.024 | 0.030 |
| | AMM | 0.012 | 0.011 | 0.014 |
| | TSA | 0.006 | 0.006 | 0.007 |
| | Sahel P | 0.004 | 0.003 | 0.004 |
| | PDO | 0.002 | 0.002 | 0.002 |

- The explanations of Beeswarm plots can be expanded for clarity.

  *Thanks for the comment. The explanation of Beeswarm plots has been improved as suggested by the Reviewer:*

*In the SHAP beeswarm plots (Figure 9), the x-axis represents SHAP values—the impact of each feature on the clustering outcome—while the color gradient (Feature value) encodes the actual correlation value between the climatic index and SPEI-12 for each data point, ranging from low (blue) to high (red). This dual encoding enables a nuanced interpretation of the model's behavior: the position along the x-axis reflects the strength and direction of influence, while the color reveals whether strong or weak correlations drive the effect.*

- Include more information on model performance (e.g., accuracy, F1-score of RF classification for clusters).

Thanks for the comment. More details on model performance have been provided in the text:

*To evaluate the relative influence of each climatic index on the clustering process and assess the predictive performance of the classifier, we employed an explainable AI approach that integrates a Random Forest (RF) classifier with SHAP. The RF model, a robust tree-based ensemble algorithm, effectively captures complex nonlinear interactions among variables but lacks inherent interpretability. To address both model performance and transparency, a comprehensive protocol was implemented.*

*First, the dataset was split using stratified sampling into training (90%) and testing (10%) subsets to preserve the original class distribution. A Random Forest classifier (100 estimators, criterion=Gini, random_state=42) was trained on the training data, and standard evaluation metrics—accuracy, class-wise precision, recall, F1-score, and the confusion matrix—were computed on the test set. The model achieved an accuracy of 0.985 on the independent test set. Class-wise precision, recall, and F1-scores were all above 0.97, confirming the classifier's strong discriminative power (see Table S4). Second, model explainability was addressed using SHAP values computed through the TreeExplainer framework. Beeswarm plots were generated for each cluster to visualize the magnitude and direction of feature contributions. Moreover, for each cluster, mean absolute SHAP values were computed for each feature, and a bootstrap procedure (n = 100) was performed to calculate 95% confidence intervals, providing statistical robustness to the importance rankings.*

In addition, Table S4 has been provided in supplementary material, reporting the output accuracy of the Random Forest model.

**Table S4. Output accuracy of the Random Forest model. The color bar ranges from red (low values) to green (high values).**

| Clusters | precision | recall | f1-score | support |
|---|---|---|---|---|
| C1 | 1 | 0.97 | 0.985 | 33 |
| C2 | 0.964 | 1 | 0.982 | 54 |
| C3 | 1 | 0.978 | 0.989 | 46 |
| accuracy | | | 0.985 | 133 |
| macro average | 0.988 | 0.983 | 0.985 | 133 |
| weighted average | 0.985 | 0.985 | 0.985 | 133 |
| Overall Random Forest Accuracy | 0.985 | | | |

**Discussion**

- Integrate more discussion on potential policy or adaptation strategies based on cluster-specific vulnerabilities.

  Thanks for the comment. In Section 4.3 "*Advancing Hydrological Clustering: From Conventional Methods to SHAP-Enhanced Insights*" details related to adaptation strategies based on the cluster outcomes are now reported:

  *In addition, the spatial heterogeneity revealed across the three clusters highlights the need for targeted adaptation strategies that align with each cluster's specific climatic vulnerabilities. Cluster C2, which faces the most severe drought intensification, would benefit from proactive investment in water harvesting infrastructure, drought-resilient crop varieties, and transboundary water governance mechanisms to manage shared resources. Cluster C1, more strongly influenced by global warming indicators such as GMT and IPWP, may require policies focused on long-term resilience—such as promoting sustainable groundwater extraction, enhancing soil moisture retention through agroecological practices, and integrating climate-smart irrigation systems. In contrast, Cluster C3, where local and regional dynamics dominate, presents an opportunity for community-based water management, improved land use planning, and localized climate services tailored to support decision-making at the grassroots level. These differentiated strategies are crucial to building adaptive capacity in the Sahel and ensuring that resource allocation reflects both scientific insight and regional socio-environmental contexts.*

- Acknowledge limitations such as the temporal range of the data (1951–2018), and possible bias due to data resolution or missing climatic drivers.

  Thanks for the comment. In Section 4.4 "*Limitations and Future Directions*" details related to possible bias due to data resolution or missing climatic drivers are now reported:

  *Furthermore, the temporal range of the analysis (1951–2018), although selected to ensure consistency and adequate overlap among multiple climate indices, may not fully capture recent accelerations in climate change and extreme event frequency, especially post-2018. As newer datasets become available, extending the analysis to include the most recent years will be critical for capturing ongoing hydroclimatic shifts. Moreover, while the 0.5° spatial resolution of the Global SPEI Database is adequate for regional-scale assessments, it may smooth out local variations critical for decision-making at finer administrative levels. This can introduce spatial biases, particularly in areas where terrain, land use, or rainfall gradients are highly variable. Finally, despite the broad suite of 31 climate indices considered, the exclusion of potentially relevant drivers—such as dust aerosol concentrations, local vegetation indices, or land surface temperature—could limit the full explanatory power of*

*the model. Incorporating such variables in future iterations may improve the detection of drought triggers and*

*feedbacks, especially where local biogeophysical processes play a pivotal role.*

**Figures and Tables**

- Improve color consistency and legends for clarity (e.g., avoid ambiguous shades).

- Add numerical cluster centroids or representative climate patterns for each cluster.

Different figures have been revised to avoid ambiguous shades:

[Figure]

**Figure 3: Z parameter of the SK test: SPEI-12 map (a)**

[Figure]

**SPEI-12 - CLIMATIC INDICES - Correlation coefficient (r)**

- -1.00 - -0.50   ■ -0.50 - -0.25   ■ -0.25 - 0   ■ 0 - 0.25   ■ 0.25 - 0.50   ■ 0.50 - 1.00

[Figure]

**Figure 5. Maps of the correlations between SPEI-12 gridded data and the most correlated climatic indices (continue).**

[Figure]

**Figure 6. Correlation analysis between SPEI-12 for Cell 2042 and AMO.** The figure presents the time series of AMO and SPEI-12 for Cell 2042, located at the border between Chad and Sudan. Additionally, it includes a scatter plot illustrating their relationship on both a monthly scale and a five-year mean scale.

[Figure]

**Figure 7.** Correlation analysis between SPEI-12 for Cell 2319 and GMT. The figure presents the time series of AMO and SPEI-12 for Cell 2319, located at the Sahel's border in Central Sudan. Additionally, it includes a scatter plot illustrating their relationship on both a monthly scale and a five-year mean scale.

**Language and Style**

- Consider simplifying overly dense or jargon-heavy sentences (especially in the Introduction and Discussion).

- Check for consistency in the use of abbreviations (e.g., GMT vs. Global Mean Temperature) and ensure all acronyms are introduced properly.

Thanks for the comment. We have carefully revised the text, particularly in the *Introduction* and *Discussion* sections, to simplify overly dense or jargon-heavy sentences and improve overall readability. Additionally, we reviewed all abbreviations and acronyms throughout the manuscript to ensure consistent usage and that each term is properly introduced at first mention). These changes have been implemented to enhance both clarity and accessibility for a broader audience.

**Novelty and Impact**

- Emphasize more clearly in the Conclusion how the framework can be generalized to other regions beyond the Sahel.

We thank the Reviewer for this insightful suggestion. In response, we have revised the Conclusion section to more clearly emphasize the generalizability of our proposed framework. Specifically, we now highlight that the modular structure—comprising seasonal trend analysis, cross-correlation with large-scale climate drivers, and explainable machine learning via SHAP-driven clustering—can be readily adapted to diverse hydroclimatic contexts beyond the Sahel. This includes temperate, monsoonal, and arid regions where drought dynamics are governed by both local conditions and global climate teleconnections. The revised text underscores the framework's flexibility, interpretability, and potential to support data-informed drought risk assessment and adaptation strategies across geographically and climatically varied settings:

*This study presents a comprehensive framework for assessing drought variability in the Sahel by integrating trend analysis, cross-correlation, and an innovative SHAP-driven clustering approach. The analysis revealed that 57.5% of the region exhibits a significant drying trend in SPEI-12, particularly in the western and southeastern Sahel, driven by increasing temperatures and declining precipitation. Conversely, 19.3% of the region shows statistically significant wetting trends, highlighting the spatial heterogeneity of drought evolution primarily through increased evapotranspiration and reduced soil moisture availability. At a regional scale, AMO and NTA emerged as key modulators of drought variability, influencing distinct drought-prone zones. Clustering identified three major drought regimes, with Cluster C2 (western Sahel: Senegal, Mauritania, Mali) experiencing the most severe intensification (Z = -5.04).*

*The SHAP-driven clustering approach integrates a Random Forest (RF) model with SHAP values to identify distinct drought patterns across the Sahel. By quantifying the contribution of each climatic index to the clustering results, this method makes the model's decision-making process transparent and highlights the prominent influence of AMO and NTA on regional drought variability. This level of interpretability allows for a deeper understanding of the climatic mechanisms behind spatial drought patterns, offering a robust basis for designing targeted adaptation strategies.*

*Beyond its application in the Sahel, the proposed framework offers strong potential for generalization to other drought-prone regions worldwide. Its modular structure—combining seasonal trend detection, teleconnection analysis, and*

*explainable machine learning—can be readily adapted to different hydroclimatic contexts, including temperate zones, monsoonal climates, and arid environments. By incorporating local drought indices and relevant climate drivers, this methodology can support region-specific assessments while maintaining the advantages of transparency and model interpretability. As such, it provides a scalable and transferrable tool for advancing drought risk management in a changing global climate.*

*By bridging advanced statistical analysis with explainable AI techniques, this study contributes a novel and interpretable approach for understanding climate impacts on regional water security, offering actionable insights for policymakers, researchers, and resource managers well beyond the Sahel context.*

---

## Author Response (AR2)

**Reviewer 1**

The authors sincerely addressed most of my comments. Below are additional comments for the authors' consideration. Most of them are about the divergent results between correlation analysis and SHAP-driven clustering combined with RF. We would like to thank the Reviewer for the thorough review of our manuscript. We have revised the manuscript by carefully considering all the comments. During this process, we also introduced several additional changes, which we believe have further improved the quality and presentation of the work. We hope that the current version will be found suitable for publication. Below, we provide a detailed and point-by-point responses to the specific review comments.

**Major comments:**

1. It is unclear which indices play a major role in the Sahel drought. In the abstract, line 13 highlights GMT and IPWP. Yet, lines 19-20 emphasize AMO and NTA. Does it mean that the correlation analysis and RF model results are inconsistent? In addition, line 15 states that AMO has a distinct influence. It is unclear what "distinct" means. I suggest the authors specify what "positive correlation" means in physical terms. For example, it could mean when AMO is at warm phase, drought severity increases, or something like that.

We thank the Reviewer for this insightful observation. We acknowledge that the abstract may have appeared inconsistent by highlighting GMT and IPWP in one part and AMO and NTA in another. We have revised the abstract and discussion to explicitly state that these approaches are not inconsistent but rather provide different perspectives: GMT and IPWP act as overarching amplifiers of drought intensity, while AMO and NTA govern the spatial differentiation of drought patterns:

Correlation analysis revealed strong negative relationships between SPEI-12 and GMT (up to -0.76) and IPWP (-0.71), underscoring their role in drought intensification. Conversely, AMO (0.40) showed a positive correlation, meaning that during its warm phase rainfall tends to increase, alleviating drought severity, while its cold phase intensifies drought. This reflects a spatially heterogeneous influence distinct from the consistently negative effects of GMT and IPWP. Using the SHAP-driven clustering, AMO and NTA emerged as key discriminators of regional drought regimes. Thus, correlation analysis and RF/SHAP highlight complementary perspectives: parameters such as GMT and IPWP drive overall drought intensification, while parameters such as AMO and NTA govern the regional differentiation of drought patterns.

2. Similar as above, line 401 indicates that AMO is the most influential feature across all three clusters, yet Fig. 5 shows AMO has weak correlation with SPEI-12 across much of the domain (most  $|\mathbf{r}| \le 0.25$ ). Conversely, indices such as

GMT and IPWP show stronger correlations (>0.25) but are not picked by RF. Please discuss why these diagnostics differ. This will better link various components of the paper.

We appreciate the Reviewer's careful comparison of the correlation maps (Fig. 5) with the SHAP-based feature importance results. The apparent discrepancy arises because the two approaches capture different aspects of the climate–drought relationship:

- Correlation analysis measures the strength of direct, linear associations between each climatic index and drought severity (SPEI-12). From this perspective, indices such as GMT and IPWP show stronger average correlations because they act as broad-scale amplifiers of drought through warming-induced evapotranspiration and moisture deficits.
- SHAP-driven Random Forest analysis, in contrast, does not rely on linearity. Instead, it evaluates how much each index contributes to distinguishing between clusters of drought regimes. Even if an index like AMO shows weak average correlation values across the entire Sahel, its spatially heterogeneous effects (positive in some areas, negative in others) and its interaction with other indices allow it to play a disproportionately large role in separating the clusters. This is why AMO emerges as a top SHAP contributor despite its modest mean correlation. Conversely, GMT and IPWP, though strongly correlated overall, do not necessarily improve the model's discrimination among clusters because their influence is more spatially uniform.

We have clarified this distinction in the revised Discussion (Section 4.1), emphasizing that correlation highlights direct associations, whereas SHAP/RF reveals the indices most relevant for identifying distinct hydroclimatic regimes. These diagnostics should therefore be viewed as complementary rather than contradictory:

Beyond linear associations, the correlation structure between global climate drivers and regional drought variability reveals an intricately woven network of teleconnections that challenge traditional dichotomies of cause and effect. The AMO emerges as a bifurcated influence, exerting positive correlations in western and central-eastern Sahel but negative correlations in central-western regions such as Burkina Faso. This spatially divergent response suggests that the AMO does not exert uniform control over Sahelian drought but rather interacts with localized boundary conditions in ways that defy simplistic interpretations. It is important to note that the apparent differences between the correlation maps (Fig. 5) and the SHAP-based feature importance results do not represent inconsistencies but rather reflect the different nature of these diagnostics. Correlation analysis highlights the strength of direct, linear associations between climatic indices and drought intensity, which explains why GMT and IPWP show strong correlations across much of the Sahel. In contrast, SHAP-driven Random Forest analysis evaluates the contribution of each index to the classification of distinct drought regimes. Although AMO shows relatively weak average correlations, its spatial heterogeneity and nonlinear interactions with other indices allow it to emerge as a key

discriminator among clusters. Conversely, the more spatially uniform influence of GMT and IPWP, while important for overall drought intensification, contributes less to distinguishing regional drought regimes. Thus, correlation and SHAP/RF provide complementary insights: the former identifies direct associations with drought severity, while the latter uncovers the indices most relevant for separating hydroclimatic regimes. Similarly, GMT and IPWP exhibit strong negative correlations with SPEI-12 (-0.76 and -0.71), reinforcing their role as primary drought intensifiers, while Sahel P maintains a positive correlation (0.22), acting as a partial counterbalance to the prevailing drying trend. The weaker and inconsistent influence of AO and NAO underscores the selective and spatially constrained nature of extratropical climatic influences on the Sahel.

3. It is unclear based on what criteria that the authors determine the "most correlated climatic indices" (Fig. 5). If the authors used the mean correlation, AMO, CAR, and TNA actually have low absolute means (0.06, -0.07, -0.07), while indices that are not shown, such as Niño-4, TNI, and Niño-3.4 have larger absolute means (-0.20, 0.15, -0.12). If instead the authors used extremes (e.g., minimum/maximum values), please justify why a single extreme value is an appropriate criterion. Also note that mixing positive and negative correlations can lead to a small mean value even when |r| is large. I recommend reporting the mean of the absolute correlations to better reflect the correlation strength. We thank the Reviewer for this helpful comment. In fact, Figure 5 reports a selection of the most correlated climatic indices rather than an exhaustive list. Based on the mean of the absolute correlations, the eight indices with the strongest overall relationships to SPEI-12 are IPWP, TSA, GMT, PDO, Sahel P, Niño-4 (now added to Figure 5), NTA, and WHWP.

However, we also chose to include additional indices such as AMO, AMM, CAR, and TNA because, although their mean correlations were relatively low, they exhibited distinctive spatial patterns in certain parts of the Sahel, including strong localized positive or negative peaks. These indices therefore provide important insights into regional heterogeneity that would not be captured by absolute mean values alone.

To avoid confusion, we have revised the text to clarify this selection criterion. Specifically, the sentence "while Figure 5 provides the maps of the correlations between SPEI-12 gridded data and the most correlated climatic indices" has been updated to:

Figure 5 provides the maps of correlations between SPEI-12 gridded data and a subset of 12 climatic indices, selected either for their high mean absolute correlations (IPWP, TSA, GMT, PDO, Sahel P, Niño-4, NTA and WHWP) with SPEI-12 or for their distinctive spatial patterns across the Sahel (e.g., AMO, AMM, CAR, TNA).

The updated Figure 5 is shown below:

Figure 5. Maps of the correlations between SPEI-12 gridded data and a subset of 12 climatic indices (continue).

Table S3. Mean, maximum, minimum values, standard deviation of the correlations, and mean of the absolute correlations, calculated between SPEI-12 gridded data and climatic indices. The color bar ranges from red (negative correlations) to blue (positive correlations) and from green (lower standard deviation) to yellow (higher standard deviation).

| Correlation coefficient (r) | Mean  | Max   | Min   | Dev.st | Mean of  r |
|-----------------------------|-------|-------|-------|--------|------------|
| AMM                         | 0.12  | 0.39  | -0.34 | 0.11   | 0.13       |
| AMO                         | 0.06  | 0.40  | -0.59 | 0.16   | 0.14       |
| Arctic Osc.                 | -0.02 | 0.15  | -0.15 | 0.04   | 0.04       |
| BEST                        | -0.11 | 0.09  | -0.26 | 0.06   | 0.11       |
| CAR                         | -0.07 | 0.25  | -0.61 | 0.14   | 0.12       |
| EPO                         | 0.00  | 0.11  | -0.11 | 0.04   | 0.03       |
| GBI                         | 0.00  | 0.05  | -0.08 | 0.02   | 0.01       |
| GMT                         | -0.33 | 0.01  | -0.76 | 0.11   | 0.33       |
| IPWP                        | -0.39 | -0.08 | -0.71 | 0.10   | 0.39       |
| NAO                         | -0.03 | 0.08  | -0.12 | 0.04   | 0.04       |
| NAO (Jones)                 | 0.03  | 0.13  | -0.06 | 0.03   | 0.03       |
| Nino1+2                     | -0.07 | 0.02  | -0.14 | 0.03   | 0.07       |
| Nino-3                      | -0.11 | 0.04  | -0.21 | 0.04   | 0.11       |
| Nino3+4                     | -0.12 | 0.07  | -0.28 | 0.05   | 0.12       |
| Nino4                       | -0.20 | 0.00  | -0.37 | 0.06   | 0.20       |
| NOI                         | 0.08  | 0.20  | -0.11 | 0.05   | 0.08       |
| North Pasific               | 0.05  | 0.10  | -0.03 | 0.02   | 0.05       |
| NTA                         | -0.15 | 0.18  | -0.61 | 0.11   | 0.16       |
| ONI                         | -0.06 | 0.15  | -0.26 | 0.06   | 0.07       |
| PDO                         | -0.23 | 0.10  | -0.42 | 0.09   | 0.24       |
| PMM                         | -0.01 | 0.16  | -0.21 | 0.05   | 0.04       |
| PNA                         | -0.07 | 0.03  | -0.19 | 0.03   | 0.07       |
| QBO                         | -0.02 | 0.14  | -0.19 | 0.06   | 0.05       |
| Sahel P                     | 0.22  | 0.35  | -0.06 | 0.06   | 0.22       |
| SOI                         | 0.10  | 0.24  | -0.08 | 0.06   | 0.11       |
| Solar Flux                  | 0.05  | 0.26  | -0.22 | 0.07   | 0.07       |
| TNA                         | -0.07 | 0.23  | -0.55 | 0.12   | 0.11       |
| TNI                         | 0.15  | 0.30  | -0.04 | 0.05   | 0.15       |
| TSA                         | -0.33 | -0.05 | -0.54 | 0.08   | 0.33       |
| WHWP                        | -0.16 | 0.07  | -0.49 | 0.09   | 0.16       |
| WPI                         | 0.00  | 0.10  | -0.08 | 0.03   | 0.02       |

4. There is inconsistency across sections. Line 381 states that "In Cluster C1, the AMO, CAR and TNA emerged as the most influential variables", and line 386 states "indices like PDO, GMT, IPWP, and TSA show very limited SHAP influence". Yet line 541 describes Cluster C1 as "more strongly influenced by global warming indicators such as GMT and IPWP."

We thank the Reviewer for highlighting this inconsistency. In the Results (lines 381–386), SHAP analysis identified AMO, CAR, and TNA as the most influential variables in Cluster C1, while GMT and IPWP showed only limited SHAP influence. However, in the Discussion (line 541), Cluster C1 was mistakenly described as being strongly influenced by GMT and IPWP. We have revised the Discussion accordingly: it now emphasizes AMO, CAR, and TNA as the primary drivers in C1, while noting that GMT and IPWP exert a broader, Sahel-wide effect but not specifically on this cluster. This correction ensures consistency between the Results and Discussion:

Cluster C1, more strongly influenced by indices such as AMO, CAR, and TNA according to SHAP analysis, may require policies focused on long-term resilience, such as promoting sustainable groundwater extraction, enhancing soil moisture retention through agroecological practices, and integrating climate-smart irrigation systems. Although global warming indicators such as GMT and IPWP play an overarching role in drought intensification across the Sahel, their direct influence on Cluster C1 is comparatively limited.

**Minor comments:**

1. Lines 260–270: IPWP is reported with a mean correlation of –0.79, but the strongest negative correlation is reported as –0.71. Please double-check these values.

Thanks for the comment. Correlations have been checked, and the values have been corrected:

The GMT (IQR = 0.14, mean = -0.33) and IPWP (IQR = 0.13, mean = -0.39) indices exhibited strong negative correlations, reaching values of -0.76 and -0.71, respectively.

2. Line 188: A closing bracket is missing.

Thanks for the comment. The closing bracket has been added:

(values with RF models)